# RIVER: A REAL-TIME INTERACTION BENCHMARK FOR VIDEO LLMS

**Yansong Shi**[1,2*] **Qingsong Zhao**[3,2*] **Tianxiang Jiang**[1,2*] **Xiangyu Zeng**[4,2] **Yi Wang**[2]
**Limin Wang**[4,2,†]
[1]School of Information Science And Technology, University of Science and Technology of China
[2]Shanghai Artificial Intelligence Laboratory
[3]College of Computer Science and Artificial Intelligence, Fudan University
[4]State Key Lab of Novel Software Technology, Nanjing University
shiyansong@pjlab.org.cn    lmwang@nju.edu.cn

## ABSTRACT

Multimodal large language models (MLLMs) have demonstrated impressive capabilities, yet nearly all operate in an offline paradigm, hindering their real-time interactivity. To address this gap, we introduce the **R**eal-t**I**me int**ER**action Benchmark for **V**ideo LLMs (RIVER Bench), designed for evaluating their real-time interaction ability with humans through perceiving the streaming videos. RIVER Bench introduces a novel evaluation framework comprising Retrospective Memory, Live-Perception, and Proactive Response tasks, closely mimicking interactive dialogues with humans rather than understanding the entire videos at once. We conduct detailed annotations using videos from diverse sources and varying lengths, and precisely defined the real-time interactive format. Evaluations across various model categories reveal that while offline models perform well in single question-answering tasks, they struggle with real-time processing. Addressing the limitations of existing models in online interaction paradigm, especially their deficiencies in long-term memory and future perception, we proposed a general improvement method that enhances models' flexibility in real-time interaction. We believe this work will significantly advance the development of real-time interactive video understanding models and inspire future research in this emerging field. Datasets and code are publicly available at https://github.com/OpenGVLab/RIVER

## 1 INTRODUCTION

The emergence of online multimodal interaction, where AI systems process streaming visual inputs while maintaining temporal-aware dialogues, presents new challenges for evaluating multimodal large language models (MLLMs). Although models like GPT-4o (Achiam et al., 2023) and Gemini (Team et al., 2024) demonstrate impressive capabilities in controlled settings, existing benchmarks inadequately address the dynamic requirements of online applications such as augmented reality navigation (Cheliotis et al., 2023) or robotic task supervision (Brohan et al., 2022; Jaegle et al., 2021), creating bottlenecks for systematic progress in online interaction research.

Existing studies (Weichbroth, 2025; Li et al., 2023a) summarize this task requires a reasonable and timely response to human requests regarding historical / current / future status based on video content. Previous works such as VideoLLM-online (Chen et al., 2024a), OV-Bench (Huang et al., 2024), and OVO-Bench (Li et al., 2025) have established preliminary frameworks for these aspects. We posit that an effective online MLLM (oMLLM) must exhibit three core competencies: (1) long-term memory retention to track evolving visual narratives, (2) precise temporal grounding for timely responses to dynamic queries, and (3) proactive reasoning to anticipate future states. Current benchmarks inadequately quantify the temporal degradation of memory (e.g., forgetting curves) or the joint optimization of response accuracy and timeliness, critical factors for real-world deployment.

---

[*]Equal contribution.
[†]Corresponding author.

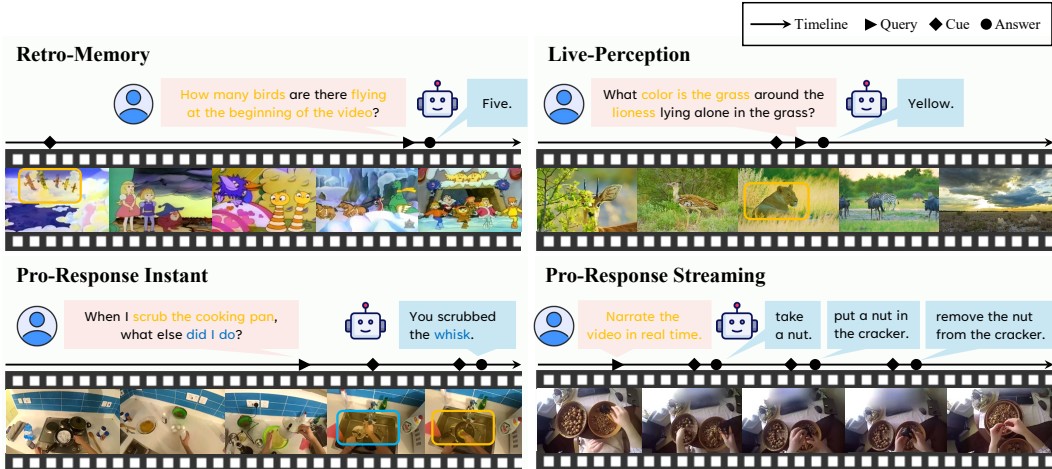

Figure 1: Illustration of different online interaction tasks. The question (Query), reference events (Cue), and answers timings are represented by ▶, ◆ and ●, respectively. Based on the frequency and timing of reference events, questions, and answers, we further categorize online interaction tasks into four distinct subclasses, as visually depicted in the figure. For the Retro-Memory, the clue is drawn from the past; for the Live-Perception, it comes from the present—both demand an immediate response. For the Pro-Response task, Video LLMs need to wait until the corresponding clue appears and then respond as quickly as possible.

Therefore, our work focuses on quantifying the retrospective and anticipatory capabilities of oMLLMs by precisely evaluating their temporal feedback. The former computes the memorization variations of oMLLM over time, and the latter reflects the timing of a prompt to a proactive plan. In addition, we also measure the response to the live multimodal understanding for formal integrity.

To measure the temporal awareness of oMLLM regarding live queries, especially in retrospection and anticipation, we design a online interaction benchmark RIVER Bench. It inherits the evaluations on the key aspects from seminal works featuring Retrospective Memory (Retro-Memory), Live-Perception and Proactive Response (Pro-Response). Specifically, in retro-memory, we measure memory persistence by analyzing performance decay over increasing temporal intervals, modeled by forgetting curve analysis. For live-perception, we quantify real-time multimodal understanding through time-sensitive question answering with latency-accuracy tradeoff optimization. Regarding pro-response, we evaluate future-state prediction via joint modeling of event forecasting confidence and temporal localization precision.

Based on a large number of existing video evaluation datasets (Vript-RR (Yang et al., 2025), LVBench (Wang et al., 2024a), LongVideoBench (Wu et al., 2025), Ego4D (Grauman et al., 2022) and QVHighlights (Lei et al., 2021)), we have rigorously selected and validated the data through both manual and automated methods, reconstructing the original annotations into multiple forms of online interaction. Our benchmark meticulously defines the timing, content, and reference points of interactions, enabling comprehensive and accurate evaluation of models' online interactive capabilities.

We conduct extensive evaluations across four model categories: vanilla offline MLLMs, sliding-window adapted models, existing online MLLMs, and models fine-tuned with our proposed training paradigm. Key findings reveal that while offline models excel at single question-answering (QA) tasks by leveraging full video context, their ability to process and interpret videos in strict real-time scenarios remains severely limited. Traditional approaches adopt sliding-window techniques, trading comprehensive temporal comprehension for partial real-time responsiveness. However, recent advances in online video understanding frameworks have demonstrated marked improvements in multiple dimensions of real-time processing, including latency, accuracy, and temporal reasoning, showcasing significant advantages over conventional offline methods. To further bridge this gap, we introduce a novel training dataset specifically designed to enhance online interaction with video content. Experimental results validate that our dataset substantially improves the real-time comprehension capabilities of existing state-of-the-art models, enabling robust performance in complex, dynamic, streaming video environments.

In summary, the contributions of this paper are as follows:

- We define the interactive form of online video comprehension. Furthermore, we propose the RIVER Bench, which provides precise annotations and questions for events at different positions in videos, enabling quantitative evaluation and comparison of models' perceptual capabilities regarding past, present, and future events, as well as their real-time interactive abilities.

- Towards enabling robust long-term memory in multimodal models, we incorporate a long-short term memory module to dynamically preserve visual information. Validated across multiple architectures, this general approach represents a pivotal step towards substantially boosting temporal understanding.

- We construct a specialized interactive training dataset to meet future interactive demands. Through fine-tuning, the model's capability for future interactions is markedly improved.

## 2 RELATED WORKS

**Online Interaction.** Online interaction with video content is characterized by two capabilities: memory and anticipation, which are not adequately addressed in previous studies. Some models have explored offline long video understanding, e.g., LongVA (Zhang et al., 2024c) and VideoChat-Flash (Li et al., 2024b), achieving impressive results by extending the input frame capacity of offline models to tens of thousands of frames. However, without effective memory mechanisms, GPU memory will eventually overflow as time progresses. To address this, several models, including MovieChat (Song et al., 2024), VideoLLaMB (Wang et al., 2024d), VideoStreaming Qian et al. (2025), Flash-VStream (Zhang et al., 2024a), and VideoChat-Online (Huang et al., 2024), have incorporated memory cache modules. When new video information is input, these models store relevant information in the memory module and clear redundant data to free up valuable storage space. StreamForest (Zeng et al., 2025) uses a Persistent Event Memory Forest to adaptively merge event-level trees based on temporal, similarity, and frequency penalties, maintaining constant memory and enabling efficient long-term retention with real-time reasoning even at extreme compression.

In terms of real-time capabilities, other models such as MMDuet (Wang et al., 2024c) and VideoLLM-Online (Chen et al., 2024a) have enabled models to acquire the ability for proactive online interaction through the construction of specialized training data, allowing them to understand user intent and provide responses or guidance. Specifically, VideoLLM-Online (Chen et al., 2024a) achieves this by using the LM Head to predict special tokens, which determine when the model should respond. MMDuet (Wang et al., 2024c), on the other hand, calculates the information and relevance scores for each sampled video frame and decides whether the model (i.e., the assistant role) should interrupt the video and initiate its turn. StreamChat (Liu et al., 2024b) updates the visual context information in real-time during the decoding process, ensuring that the latest visual information is used during decoding, which is crucial for the real-time operation of online models.

Furthermore, some models, such as GPT-4o (Achiam et al., 2023), support voice input along with other modalities. Examples include IXC2.5-OL (Zhang et al., 2024b), VITA-1.5 (Fu et al., 2025), and MiniCPM-o-2.6 (Yao et al., 2024). IXC2.5-OL (Zhang et al., 2024b) also incorporates a memory module that selectively retains relevant information. However, its voice module integrates multiple components, resulting in a less streamlined design. VITA-1.5 (Fu et al., 2025) draws inspiration from the Freeze-Omni model (Wang et al., 2024b), enabling a more natural integration of voice and video understanding capabilities. Since voice interaction aligns more closely with everyday communication, it holds significant potential for deployment on edge devices.

**Online Video Benchmarks.** Traditional offline video understanding focuses on holistic comprehension of video content, and numerous excellent benchmarks (Li et al., 2024a; Fu et al., 2024) have emerged and gained widespread adoption. However, the task types and evaluation methods for online video understanding have not been precisely defined. Some models have attempted to enhance online capabilities and proposed their own benchmarks, but the evaluation formats remain largely similar to traditional offline benchmarks, failing to adequately address the requirements of online video understanding. For instance, VStream-QA (Zhang et al., 2024a), composed of VStreamQA-Ego and VStream-QA-Movie, incorporates reference timestamps in its question-answer pairs, which are crucial for online video understanding tasks. It uses GPT-4V (Achiam et al., 2023) to generate frame descriptions and GPT-4 (Achiam et al., 2023) to create multiple-choice questions, covering

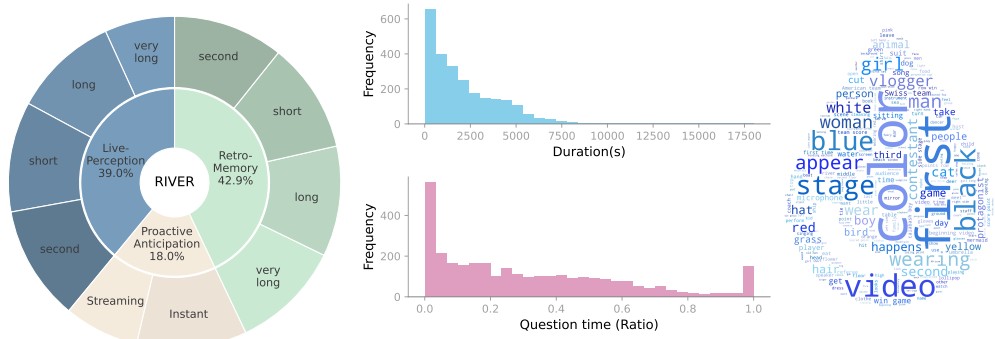

Figure 2: The pie chart on the left illustrates the quantitative distribution of various tasks within the benchmark. The two bar charts in the middle depict the statistics of video duration and the proportion of question timestamps, respectively. On the right, a word cloud which is constructed from the annotated textual data within the dataset, visually emphasizing the most frequently occurring terms.

scene and event descriptions, temporal existence, and sequence judgment. However, its evaluation format does not fully capture the interactive and real-time nature of online video understanding. Similarly, VideoLLM-Online (Chen et al., 2024a) emphasizes the fluency and timeliness of responses but neglects the accuracy of question answering, which is critical for interactive systems.

MovieChat-1K (Song et al., 2024) introduces a breakpoint mode, which involves asking questions at specified timestamps in the video stream, aligning with the immediate response type in online video understanding. However, it lacks a comprehensive framework for other interaction types, such as recalling or awaiting responses. StreamingBench (Lin et al., 2024) explicitly links questions to specific timestamps and incorporates multi-modal information and complex contextual details from the video stream, addressing tasks like real-time visual understanding, omni-source understanding, and contextual understanding. Despite its rich task design, it does not fully formalize the dynamic nature of online interactions. OV-Bench (Huang et al., 2024) defines three temporal categories—current, past, and future—based on the relationship between the question timestamp and the video timeline, transforming tasks like grounding and STAL (Spatio-Temporal Action Localization) into an online understanding format. While it introduces temporal categorization, it does not explicitly address the continuous and interactive aspects of online interaction tasks. OVO-Bench (Li et al., 2025) categorizes online video understanding tasks into three types: Backward Tracing, Real-Time Visual Perception, and Forward Active Responding, featuring a flexible format for questions and answers. It represents the most relevant existing work for defining online video understanding tasks but it lacks fine-grained temporal segmentation of the response or clue intervals, which is crucial for analyzing the online interaction capability of Video LLMs. In contrast, our proposed RIVER Bench provides a more comprehensive and precise definition of online interaction with video, encompassing retro-memory, live-perception, and pro-response. This approach addresses the limitations of existing benchmarks and offering a robust evaluation standard for future research.

## 3 RIVER BENCH

We design RIVER Bench for assessing the temporal awareness of MLLMs on online interactions. It measures the retrospective memory, live perception, and proactive anticipation of oMLLMs by evaluating the accuracy, relevance, and timeliness of their generated responses. Based on the temporal relationships between the moments of clues, questions, and answers, we categorize the questions in the benchmark into three main types, as shown in Figure 1. The statistic results of RIVER Bench are shown in Table 1 and Figure 2. We first present the formulation, followed by a detailed description of the benchmark construction and its characteristics.

**Formulation.** We formulate this online interaction into a window-based video-text-to-text task, as:

$$\mathcal{L} = -\log P_\theta(r_t | \mathbf{V}_{t':t}, q, h_{<t'}, r_{<t}),$$

Table 1: Comparison of RIVER and other video online benchmarks.

| Benchmark | Videos | Questions | Memory & Perception | | | | | Anticipation | |
| --- | --- | --- | --- | --- | --- | --- | --- | --- | --- |
| | | | General | Short | Medium | Long | Very Long | Instant | Stream |
| VStream-QA (Zhang et al., 2024a) | 32 | 3,500 | ✓ | ✗ | ✗ | ✗ | ✗ | ✗ | ✗ |
| StreamingBench (Lin et al., 2024) | 900 | 4,500 | ✓ | ✗ | ✗ | ✗ | ✗ | ✗ | ✗ |
| OVBench (Huang et al., 2024) | 1,463 | 4,874 | ✓ | ✗ | ✗ | ✗ | ✗ | ✗ | ✗ |
| OVO-Bench (Li et al., 2025) | 644 | 3,100 | ✓ | ✗ | ✗ | ✗ | ✗ | ✓ | ✓ |
| **RIVER (ours)** | 1,067 | 4,278 | ✓ | ✓ | ✓ | ✓ | ✓ | ✓ | ✓ |

where $P$ denotes the oMLLM parametrized by $\theta$. $\mathbf{V}_{t':t}$ is the streaming video started at time $t'$ until now $t$ , while $q$, $r$, $h_{<t}$ stand for users' queries (in words), desired responses, and the historical modeling of $P$. Note for each $r$ in learning, it can contain more than one EOS (end-of-sequence) tokens at arbitrary locations for simulating silence or pauses in a live conversation.

## 3.1 INTERACTIVE TASK TYPES

We summarize three main task types of RIVER Bench as retro-memory, live-perception, and pro-response, according to the happening time (denoted as $t_{\mathbf{V}}$) of the queried event or target.

- Retro-Memory: oMLLM answers users based on the history ($h_{<t'}$), as users' question concerning past events (where $t_{\mathbf{V}} < t'$), such as "Where did I just put my bag?".
- Live-Perception: oMLLM responds immediately as the question relates to the current or short-term visual inputs (where $t' \leq t_{\mathbf{V}} \leq t$).
- Pro-Response: For this task, the oMLLM must continuously monitor the video stream and execute a response precisely when a user-specified condition is met (where $t_{\mathbf{V}} > t$). For example, when the user is searching for a wrench that is temporarily not visible, once the wrench appears, oMLLM will inform the user. A special case involves the model continuously narrating its visual inputs upon request (e.g., 'keep describing what you see'), a task analogous to dense video captioning (Johnson et al., 2016; Krishna et al., 2017).

Note previous works have already defined these types and evaluated them in forms. Compared with seminal researches (Li et al., 2025; Huang et al., 2024), we intend to study MLLM's understanding curve of historical and future events according to users' queries. Specifically, how the time interval $\Delta = \|t_{\mathbf{V}} - t\|$ ($\Delta = 0$ when $t' \leq t_{\mathbf{V}} \leq t$) impacts oMLLM's performance on retro-memory and pro-response.

## 3.2 DATA CONSTRUCTION

Unlike conventional QA benchmarks, online video comprehension benchmarks emphasize inter-activity, requiring precise definitions of query time, cue time, and response time. To this end, we meticulously curate datasets from diverse sources, ensuring each question is grounded in accurate temporal annotations that explicitly specify these time points. Specifically, we utilized datasets from Vript-RR (Yang et al., 2025), LVBench (Wang et al., 2024a), LongVideoBench (Wu et al., 2025), Ego4D (Grauman et al., 2022) and QVHighlights (Lei et al., 2021), which underwent rigorous filtering, restructuring and verification to form the final version of the benchmark. The process of data processing and generation is presented in Figure 3.

**Retro-Memory.** For retro-memory questions, we evaluate the model's ability to recall events across different time spans by categorizing queries into four distinct temporal intervals: short (15-30s), medium (30-60s), long (300-900s), and very long (1800-3600s). To ensure precise evaluation, each question is carefully designed so that its correct answer can only be derived from one specific moment in the video. We constructed this benchmark by systematically filtering and restructuring datasets from multiple sources, including Vript-RR (Yang et al., 2025), LVBench (Wang et al., 2024a), and LongVideoBench (Wu et al., 2025), inserting questions at strategically selected temporal positions. The resulting evaluation set consists of 1.5k high-quality multiple-choice QA pairs, each with precisely annotated timestamps for the queried events to eliminate any temporal ambiguity in both questions and answers.

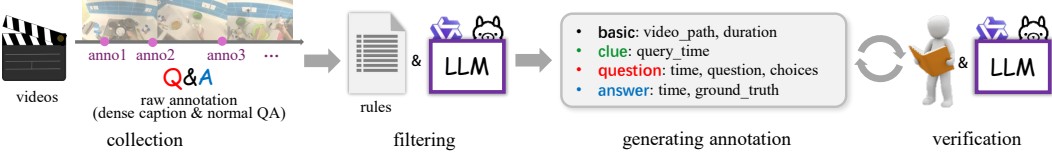

Figure 3: Illustration of data processing process.

**Live-Perception.** Live-Perception questions assess a model's ability to comprehend visual information within the current temporal window (typically a few seconds), which is also a fundamental capability for video multimodal models. The queries encompass environmental context, actions, object attributes (e.g., quantity, color), with particular focus on both dynamic sequences of changes and rich static visual semantics. The live-perceptionquestion remains similar to conventional video understanding tasks, but emphasises real-time interaction. We derived these questions from the same data sources as the past-oriented questions, totaling 0.4k samples.

**Pro-Response.** For pro-response questions, we categorize them into instant and stream types, representing single-answer and multi-answer scenarios, respectively. Stream-type questions require the model to continuously describe the scene or guide the user, generating textual outputs at different timestamps. While similar to dense captioning tasks, this process operates in a real-time conversational manner rather than post-video analysis. Following the data construction approach of VideoLLM-online, we curated evaluation data for continuous responses from the Ego4D-Narration validation set, totaling 1.2k samples. Instant-type questions arise when current or past video information is insufficient for accurate recall or retrieval, requiring the model to observe further and respond at the appropriate moment (e.g., predicting the next event). In our benchmark, the model must not only determine the optimal timing but also provide coherent answers. We filtered raw annotations from Ego4D-Narration and QVHighlights, which cover both egocentric and third-person videos, using sentence embeddings (Devlin et al., 2019), then synthesized anticipatory questions and distractor options via LLMs (Yang et al., 2024). Based on temporal intervals, instant-type questions are further classified into short, medium, long, and very long, totaling 1.4k samples.

### 3.3 QUALITY CONTROL

**Removing low-quality question-answer pairs.** To ensure high-quality question-answer pairs in our benchmark, we employ a multi-stage filtering process combining open-source large language models (LLMs) (Yang et al., 2024; AI, 2024) and rigorous human evaluation. First, we use LLMs to identify and remove questions that can be answered correctly without visual input, thereby mitigating the influence of language priors and model biases. Questions that are overly broad, span excessively long time durations, or rely on ambiguous cues are also filtered out, as they may be answerable with minimal visual context and do not adequately test a model's recall or temporal reasoning abilities.Detailed guidelines and examples of our review criteria are provided in Appendix A.

**Removing ordinary and meaningless event descriptions.** To reduce reliance on purely temporal queries and instead emphasize behavior prediction and event sequencing grounded in contextual localization, we use semantic-similarity metrics to curate annotations. This process isolates distinctive events that serve as unambiguous reference anchors for questions. Each anchor is paired with a precise timestamp, furnishing clear yet concise temporal context. The resulting dataset minimizes ambiguity and provides a rigorous test of a model's capacity to recall video content accurately.

### 3.4 METRICS

**Retro-Memory & Live-Perception.** After obtaining model responses, we first use regular expression matching to extract answers for multiple-choice (MC) questions. For failed matches or non-MC questions, we adopt an open-ended (OE) evaluation approach following (Song et al., 2024). We leverage the open-source Qwen2.5-72B (Yang et al., 2024) to assess response consistency with reference answers. To ensure a fair comparison, we have presented the accuracy rates of both methods (MC & OE) as comprehensively as possible.

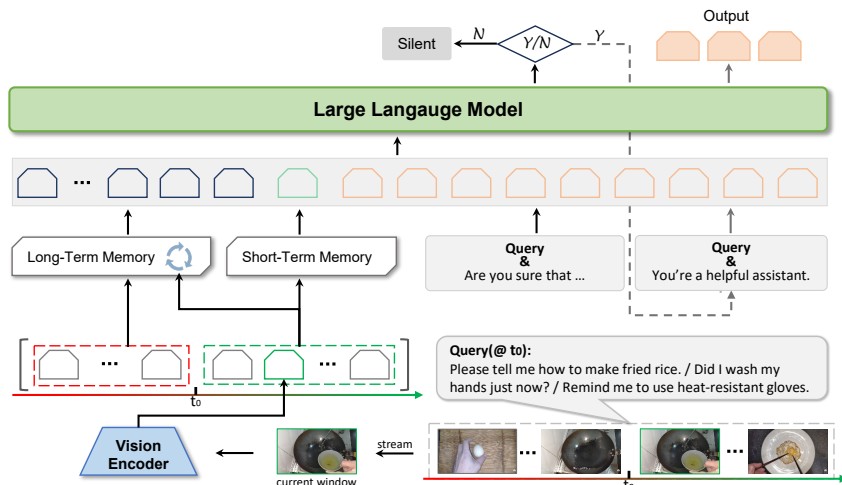

Figure 4: Pipeline to enable MLLMs to support online inference capabilities. The Long Short-Term Memory module continuously receives new visual features and selects the most important parts. After a query is posed at $t_0$, the model is queried at each time window; if it decides to answer, the final response is output.

**Pro-Response.** To evaluate proactive response capabilities, we introduce a Response Accuracy Metric that quantifies performance based on temporal alignment with ground-truth timestamps $t_g$. Defined within a tolerance window of length $w$ centered at $t_g$, the metric assigns a full score to responses falling inside this interval, reflecting acceptable anticipation. To align with human subjective tolerance, the scoring function strictly penalizes early responses with a score of zero to discourage false alarms, while applying a linear decay to late responses. Specifically, scores for delayed predictions gradually decrease to zero as the latency increases beyond the window, acknowledging their diminishing utility without immediately discarding them as total failures.

## 4 EXPERIMENTS

Utilizing the proposed RIVER Bench, we evaluate four categories of video-processing multimodal large language models: (1) commercial closed-source models, (2) open-source models with native online inference support, (3) open-source video multimodal models, and (4) our proposed video multimodal model adapted for online inference.

To enable these models to handle ultra-long videos (up to 120 minutes), we applied their respective recommended frame sampling strategies. For GPT-4o Achiam et al. (2023) and Gemini-1.5-pro (Team et al., 2024), we extracted 50 frames; for VideoChat2 (Li et al., 2024a), LLaVA-Video (Zhang et al., 2024d), and InternVL2.5 (Chen et al., 2024b), we extracted 16 frames each. For the two models with native online inference support, VideoLLM-Online (Chen et al., 2024a) and Flash-VStream (Zhang et al., 2024a), we processed videos at 4 frames per second (fps) before feeding them into the models.

For each retro-memory and live-perception question, we used the following prompt as model input: $<$ System Prompt $><$ Video $><$ VideoHere $>< /Video >$ *Based on your observations, select the best option that accurately addresses the question. Question. A) Option1; B) Option2; C) Option3; D) Option4. Only give the best option.*

For pro-response question, we provided the models with the following prompt as input: $<$ System Prompt $><$ Video $><$ VideoHere $>< /Video >$ *Based on your observations, follow the instructions carefully and explain your answers.*

### 4.1 MAKING OFFLINE MODELS WORK ONLINE

Typical multimodal large language models process the entire video input before answering user questions. This inference paradigm inherently limits their ability to handle online interactive tasks.

Table 2: RIVER Bench evaluation results regarding the core online video understanding capability. "#F" indicates the total number of frames processed by the model during inference. "Loc" measures how precisely the model's response falls within the correct time window in Pro-Response task. "∅" means that some models do not have the corresponding capabilities. All numbers in the table are presented in percentage (%). The gray background indicates methods that either natively support online inference or have been adapted to enable online inference capabilities.

| Models | LLMs | #F | Retro-Memory | | Live-Perception | | Pro-Response | | |
|---|---|---|---|---|---|---|---|---|---|
| | | | | | | | Instant | | Streaming |
| | | | OE | MC | OE | MC | Loc | MC | OE |
| *Closed-Source Models* | | | | | | | | | |
| GPT-4o | - | 50 | 39.09 | 59.56 | 40.08 | 61.05 | ∅ | ∅ | 1.63 |
| Gemini-1.5-pro | - | 50 | 24.24 | 36.35 | 34.53 | 52.19 | ∅ | ∅ | 1.51 |
| *Open-Source Models* | | | | | | | | | |
| VideoChat2 | Mistral-7B (Jiang et al., 2023) | 16 | 19.41 | 31.97 | 18.76 | 29.28 | ∅ | ∅ | 2.21 |
| InternVL2.5 | InternLM-8B (Cai et al., 2024) | 16 | **25.07** | 42.68 | **29.34** | **43.65** | ∅ | ∅ | 3.60 |
| Llava-Video | Qwen2-7B (Yang et al., 2024) | 16 | 24.94 | **46.00** | 27.57 | 41.00 | ∅ | ∅ | **4.25** |
| VideoChat-Flash | Qwen2-7B (Yang et al., 2024) | 16 | 21.91 | 41.51 | 27.91 | 41.44 | ∅ | ∅ | 3.68 |
| VideoChat2 | Mistral-7B (Jiang et al., 2023) | 1fps | 19.65 | 35.52 | 23.66 | 41.16 | 4.51 | 28.98 | 4.58 |
| InternVL2.5 | InternLM-8B (Cai et al., 2024) | 1fps | 21.13 | 39.72 | **34.68** | **58.84** | 17.95 | 30.86 | 4.09 |
| Llava-Video | Qwen2-7B (Yang et al., 2024) | 1fps | 24.51 | 42.71 | 33.87 | 51.38 | 19.50 | 27.55 | **6.21** |
| VideoChat-Flash | Qwen2-7B (Yang et al., 2024) | 1fps | **25.68** | **45.75** | 33.60 | 56.35 | **20.24** | **35.90** | **6.21** |
| Flash-VStream | Vicuna-7B (Chiang et al., 2023) | 1fps | 10.43 | 27.28 | 12.43 | 29.28 | - | - | 1.31 |

As illustrated in Figure 4, we propose a framework that integrates a sliding window sampling strategy with a long-short term memory module, enabling the models to support online inference capabilities, while simultaneously maintaining coherent temporal awareness across extended video sequences.

We employ a sliding window approach with a sampling rate of 1 frame per second (fps) for processing long video inputs. The window length is set to the recommended optimal number of frames for the models. The short-term memory consists of video frame tokens from the current window. The long-term memory module comprises compressed tokens from video frames prior to the current window, with a fixed length of $M$ memory slots. Each memory slot contains the same number of tokens as the short-term memory. In our implementation, we maintained the original optimal configurations for all models and kept the number of added long-term memory tokens consistent to ensure a fair comparison. Since VideoChat2-HD (Li et al., 2024b) employs a Q-Former (Li et al., 2023b) as the connector between visual features and the LLM, which generates a fixed number of tokens per slot, we adjusted the number of slots instead to align the total count of memory tokens. Humans tend to integrate events occurring within adjacent time intervals and abstract them into higher-level semantic representations (Nader & Einarsson, 2010). Inspired by this, we maintain long-term memory slots using a nearest-neighbor averaging strategy.

During user query inference, we explicitly equip the model with timeline information by synthesizing context from both short- and long-term memory directly into the system prompts, which includes precise timestamps and key visual features. Specifically, we use the following prompts: *The following video tokens contain a long memory of 0.0 to {:} seconds. $< Video >< Long\text{-}Term >< /Video >$ The following video tokens contains a short memory sampled from {:} to {:} seconds. $< Video >< Short\text{-}Term >< /Video >$.*

Table 3: Online evaluation results regarding the Pro-Anticipation capability.

| Models | #F | Pro-Response | | |
|---|---|---|---|---|
| | | Instant | | Streaming |
| | | Loc | MC | OE |
| Flash-VStream | 1fps | - | - | 1.31 |
| VideoLLM-Online | 2fps | 23.88 | 6.67 | 4.41 |
| VideoLLM-Online$_{+RIVER}$ | 2fps | 33.28 | 9.84 | 5.03 |
| VideoLLM-Online$_{+RIVER}$ | 4fps | **35.16** | **10.53** | **5.47** |

## 4.2 TRAINING THE ONLINE MODELS.

We adopt an architecture aligned with VideoLLM-Online (Chen et al., 2024a), comprising three core components: a visual encoder, an MLP projection layer, and a large language model. Specifically,

Table 4: Evaluation results of Retro-Memory type questions.

| Models | LLMs | #F | Short | | Medium | | Long | | Very Long | |
|---|---|---|---|---|---|---|---|---|---|---|
| | | | MC | OE | MC | OE | MC | OE | MC | OE |
| *Closed-Source Models* | | | | | | | | | | |
| GPT-4o | - | 50 | 63.26 | 41.99 | 63.26 | 41.99 | 58.01 | 38.12 | 52.21 | 34.25 |
| Gemini-1.5-pro | - | 50 | 41.77 | 28.45 | 36.98 | 24.03 | 34.73 | 23.20 | 31.94 | 20.99 |
| *Open-Source Models* | | | | | | | | | | |
| VideoChat2 | Mistral-7B (Jiang et al., 2023) | 16 | 32.04 | 19.34 | 32.32 | 20.44 | 31.49 | 19.34 | 32.04 | 18.51 |
| InternVL2.5 | InternLM-8B (Cai et al., 2024) | 16 | 46.13 | **29.28** | 47.51 | **26.52** | 39.78 | 24.03 | 37.29 | **20.44** |
| Llava-Video | Qwen2-7B (Yang et al., 2024) | 16 | **49.17** | 29.01 | **48.07** | 25.97 | **44.20** | **24.59** | **42.54** | 20.17 |
| VideoChat-Flash | Qwen2-7B (Yang et al., 2024) | 16 | 44.20 | 22.93 | 43.92 | 22.93 | 40.06 | 22.38 | 37.85 | 19.38 |
| VideoChat2 | Mistral-7B (Jiang et al., 2023) | 1fps | 34.25 | 20.44 | 36.19 | 21.27 | 33.43 | 17.13 | 32.60 | 15.75 |
| InternVL2.5 | InternLM-8B (Cai et al., 2024) | 1fps | 37.02 | 22.10 | 39.50 | 21.27 | 32.60 | 15.47 | 30.66 | 12.16 |
| Llava-Video | Qwen2-7B (Yang et al., 2024) | 1fps | **44.20** | 24.31 | 43.65 | 24.59 | 37.02 | 19.89 | 37.29 | **19.89** |
| VideoChat-Flash | Qwen2-7B (Yang et al., 2024) | 1fps | 43.92 | **24.86** | **48.90** | **29.28** | **41.44** | **22.10** | **38.12** | 18.55 |
| VideoLLM-Online | LLaMA3-8B (AI, 2024) | 2fps | - | 3.87 | - | 3.59 | - | 3.87 | - | 3.32 |
| Flash-VStream | Vicuna-7B (Chiang et al., 2023) | 1fps | 28.73 | 10.50 | 25.97 | 9.39 | 27.90 | 9.12 | 26.52 | 12.70 |

we employ the SigLIP-Large-Patch16 encoder (Zhai et al., 2023) coupled with a two-layer MLP connector to extract video frame representations at a rate of 4 frames per second (fps), following the sampling strategy of LLaVA-1.5 (Liu et al., 2024a). Each video frame is encoded as a tensor of shape $(1 + 3 \times 3) \times d$, where $d$ denotes the hidden dimension, the $3 \times 3$ term corresponds to spatially average-pooled patch tokens, and the 1 represents the global CLS token. These visual embeddings are interleaved with language tokens and jointly processed by the LLM. To enable efficient adaptation, we integrate Low-Rank Adaptation (LoRA) (Hu et al., 2022) into all linear layers of the LLaMA3-8B backbone (AI, 2024).

Consistent with VideoLLM-Online, our optimization objective combines standard language modeling (LM) loss with a streaming-specific loss to enhance temporal responsiveness. However, distinct from their setup, we train for a single epoch with a learning rate of $3 \times 10^{-5}$, utilizing DeepSpeed (Rasley et al., 2020) with ZeRO-2 optimization to maximize memory efficiency and training throughput. Notably, our training data incorporates randomized timestamps for user queries rather than anchoring them at the default 0-second mark (see Section 3.2), a design choice that significantly bolsters generalization across diverse real-world interaction scenarios.

### 4.3 EVALUATION RESULTS AND ANALYSIS.

We evaluate various multimodal large language models (MLLMs) on the RIVER Bench, assessing their core online interaction capabilities. Table 2 compares native online inference models and enhanced non-native MLLMs. GPT-4o achieves the best performance, excelling in live-perception, retro-memory, and pro-response tasks.

Non-native models augmented with our online inference pipeline deliver highly competitive results. For live-perception questions, our approach outperforms native multimodal inference. In retro-memory questions, VideoChat-Flash, fine-tuned on long videos, and InternVL2.5, trained primarily on image-text data, exhibit divergent trends. We attribute this to VideoChat-Flash's long-video compression design, which reduces sensitivity to memory token compression, while InternVL2.5 suffers performance drops without this mechanism.

Intriguingly, existing models that claim support for streaming video QA underperform substantially on our benchmark. We attribute this underperformance to two primary factors. First, Flash-VStream is optimized for long-video comprehension rather than interactive QA. Second, VideoLLM-Online's training relies on an offline methodology, similar to LLaVA-1.5, which makes it susceptible to overfitting on specific scenes and predefined QA formats, limiting its adaptability to dynamic streams. Fine-tuning VideoLLM-Online on our RIVER Bench's pro-response training set improves results (Table 3), demonstrating our data's effectiveness for enhancing online interaction, particularly in proactive response tasks (11.28% accuracy improvement over baseline).

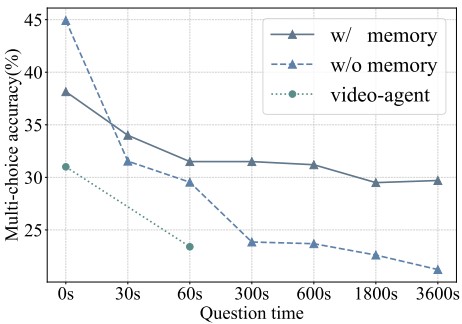

Figure 5: Memory curve of MLLM (Li et al., 2024a) and video agent (Fan et al., 2024) under different query time conditions.

Table 5: The evaluation results for different types of visual cues. FC: fine-grained visual cue. CC: causal cue. BC: background cue.

| Models | LLMs | #F | FC | CC | BC |
| | | | MC | MC | MC |
|---|---|---|---|---|---|
| VideoChat2 | Mistral-7B | 16 | 30.23 | 29.69 | 38.69 |
| InternVL2.5 | InternLM-8B | 16 | 48.74 | 37.36 | 46.23 |
| Llava-Video | Qwen2-7B | 16 | **53.16** | 38.25 | 47.04 |
| VideoChat-Flash | Qwen2-7B | 16 | 47.55 | 36.44 | 44.92 |
| VideoChat2 | Mistral-7B | 1fps | 36.06 | 31.15 | 46.89 |
| InternVL2.5 | InternLM-8B | 1fps | 42.51 | 35.29 | 46.56 |
| Llava-Video | Qwen2-7B | 1fps | 50.08 | 34.25 | 51.48 |
| VideoChat-Flash | Qwen2-7B | 1fps | 50.39 | **40.92** | **54.10** |

### 4.3.1 MODEL MEMORY CAPABILITY

Table 4 shows performance on retro-memory questions across different durations. As recall duration increases, most models exhibit declining visual memory retrieval and reasoning abilities. Flash-VStream is an exception. While its overall performance remains modest, it maintains consistent accuracy across all durations. This result underscores the criticality of memory in MLLMs, demonstrating that the choice of memory mechanism significantly impacts long-term information retrieval performance. Notably, models modified for online inference show improved performance on medium-to-long-term memory questions.

### 4.3.2 MODEL MEMORY CURVE

Figure 5 shows accuracy curves for recall-type questions across durations. Adding memory modules significantly boosts retrieval, cutting the performance drop-off (decay slope) by 12% compared to models without memory. Notably, in contrast to the classic Ebbinghaus forgetting curve (Hu et al., 2013), MLLMs equipped with these modules exhibit superior retention stability within 1-hour timeframes, suggesting fundamentally different memory mechanisms from human cognition.

### 4.3.3 PERFORMANCE ACROSS DIFFERENT CLUE CATEGORIES

Table 5 shows performance on retro-memory questions categorized by the type of visual cues: Fine-grained Cues (FC), which target specific object attributes or detailed appearances; Causal Cues (CC), which require reasoning about event dynamics and temporal dependencies; and Background Cues (BC), which focus on static scene context or environmental settings. All methods perform poorly on CC questions, revealing their greater difficulty and highlighting the need for future work on visual perception integrated with event attribution. The modified VideoChat-Flash shows improvements across all cue types, achieving best performance on context-dependent questions.

## 5 CONCLUSION

This paper addresses the challenges in evaluating multimodal large language models for online multimodal interaction. We propose RIVER Bench, which quantifies key aspects like retrospective memory, live-perception, and proactive response. Through a comprehensive evaluation of different model categories, our benchmark effectively identifies their key limitations in real-time interaction. Moreover, a specialized fine-tuning dataset is introduced. After training on this dataset, existing models show significant improvements in online video understanding, enabling better performance in dynamic, real-time scenarios.

**Limitations and Future Work.** Currently, our dataset does not include audio data. Given that sound is one of the most readily available modalities for real-time interaction, integrating audio into the evaluation of online video content is crucial. In the future, we plan to update our video annotation data to incorporate audio.

## ACKNOWLEDGEMENTS

This work is supported by the Basic Research Program of Jiangsu (No. BK20250009), the National Key R&D Program of China (No. 2022ZD0160900) and the Collaborative Innovation Center of Novel Software Technology and Industrialization.

This work was completed at the Shanghai Artificial Intelligence Laboratory. We gratefully acknowledge the computational resources provided by the Shanghai Artificial Intelligence Lab, which made this research possible.

## ETHICS STATEMENT

Our work involves the collection and annotation of videos sourced from publicly available platforms and existing open datasets. All data are selected in accordance with their respective licenses and terms of use, and we do not include any private, sensitive, or personally identifiable information. The benchmark focuses on general video understanding and interaction scenarios rather than sensitive human activities, and no clinical, offensive, or invasive content is included. Annotation was conducted by trained annotators with clear guidelines to ensure accuracy, consistency, and fairness. Potential risks such as misuse for generating misleading real-time interactions or unauthorized surveillance are mitigated by releasing the dataset strictly for academic research purposes. We adhere to the ACM Code of Ethics in all aspects of this work, including dataset documentation, annotation transparency, and clear reporting of model limitations.

## REPRODUCIBILITY STATEMENT

We have taken several steps to ensure the reproducibility of our benchmark and experimental findings. Sec. 3 and Appendix A describe the video sources, selection criteria, and dataset composition, including duration and source distribution. Sec. 3 details the construction of the real-time interaction tasks, namely Retrospective Memory, Live-Perception, and Proactive Anticipation. Evaluation metrics and detailed experimental setups for all compared models are specified in Sec. 4. All code for data processing, benchmark simulation, and evaluation, along with the benchmark data (under appropriate licenses), will be released in an anonymized repository to facilitate reproduction and extension by the research community.

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

APPENDIX

# A  IMPLEMENTATION DETAILS

## A.1  BENCHMARK DETAILS

RIVER Bench is primarily composed of two types of video data: one consisting of question-answer (QA) pairs with visual information, and the other containing only dense action annotations. The construction process of RIVER Bench is illustrated in Figure 6.

For the first category of data, we first apply a rule-based filtering process based on the existing annotations. This involves removing questions that contain specific personal names, correspond to excessively long video segments, or are overly general in nature. Subsequently, we employ a large language model (LLM)-based filtering step: if a unimodal language model can correctly answer a question without access to the visual modality, the corresponding QA pair is considered invalid.

The second category of data consists of videos annotated with dense, timestamped event descriptions. Redundant, trivial, or repetitive annotations are removed using custom-designed rules. Next, we compute sentence embeddings for each event and select the most distinctive one as the key event based on the similarity matrix. Finally, an LLM generates QA pairs using the annotation information along with a predefined question template bank. These generated QA pairs then undergo the aforementioned LLM-based filtering procedure to ensure quality.

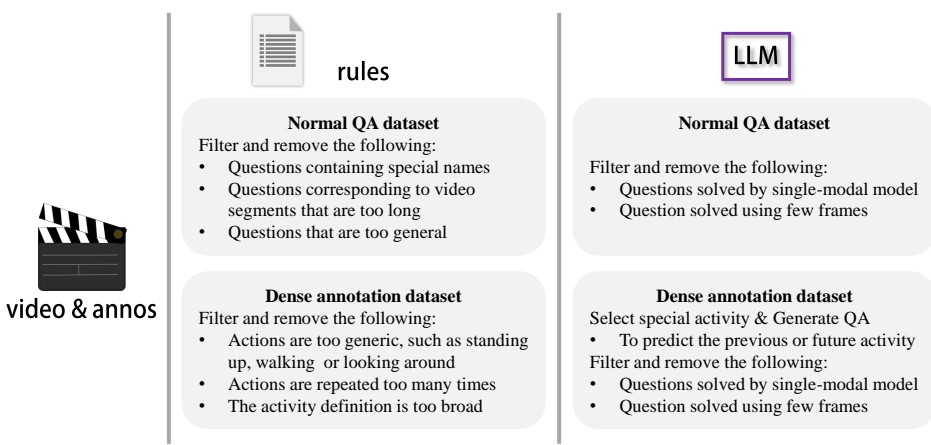

Figure 6: Illustration of data processing.

To clarify the selection process for Retro-Memory and Live-Perception questions, we used annotations from Vript-RR Yang et al. (2025), LVBench Wang et al. (2024a), and LongVideoBench Wu et al. (2025), which were filtered and reconstructed as follows:

**Vript-RR**: A subset of the Vript-Hard benchmark, designed to address ambiguity in long-video question answering. It contains 152 high-quality entries with detailed hints for locating relevant scenes and concise questions. These were directly incorporated for their quality despite the limited quantity.

**LVBench**: A benchmark for evaluating video-language models (VLMs) on extremely long videos, assessing capabilities like temporal localization, summarization, reasoning, entity recognition, event understanding, and information retrieval. We excluded questions involving specific names or those spanning entire videos to focus on temporally relevant tasks.

**LongVideoBench**: A benchmark for fine-grained multimodal retrieval and reasoning in long videos. We selected perception-related questions and removed those relying on complete dialogue as hints.

After this initial filtering, we obtain a set of valid hints and questions. We further apply multiple large language models (LLMs) to identify and remove low-quality or ambiguous instances. Finally, for

each event, we randomly sample a query timestamp after the event occurrence (for Retro-Memory) or synchronized with the event time (for Live-Perception), forming the final annotations.

During the construction of Proactive Anticipation Instant questions, the following prompt is employed. The filtered, timestamped dense event annotations are provided as input to a large language model (LLM). Based on this input, the model generates semantically relevant questions together with a correct answer and several plausible distractor options.

---

**Prompt used for generating Proactive Anticipation Instant questions**

```
You are a very intelligent multimodal assistant helps the user to
do the following tasks:
Please complete the conversation between user and assistant.  Note
that the assistant will actively provides clear, concise, real-time
language assistance.  The assistant does not know the absolute
time, so don't mention timestamp in response.  Sometimes the user
may ask irrelevant questions, the assistant is very helpful and
will also answer that.  Also, generate 3-5 wrong choices that are
easy to confuse the assistant.  Firstly, you may need to refine
the question which is asking about things that will happen in the
future.  The only one correct choice must be closely around the
current key event.  The wrong choices must contain the objects
mentioned in the key event.  All the pronouns in question and
choices should be the same and correct.
```

**Annotations:**
```
<timestamp 1>:  <annotation 1>
<timestamp 2>:  <annotation 2>
...
<timestamp n>:  <annotation n>
```
**Key event:**
```
<timestamp k>:  <annotation k>
```

**Output format**:
```
Person:  When ..., <generated question>
Correct choice:  ...
Wrong choices:  ...
```

---

The following is the template library used for synthesizing Proactive Anticipation Instant questions. Templates from this library are randomly selected to generate the questions.

---

**Questions used for Proactive Anticipation Instant**

**Next**
```
["What's the next step?", "What should I do after this?", "What
action should I take next?", "What's the next thing to do?", "What
am I supposed to do next?", "What's the next procedure?", "What's
the following procedure?", "What is to be done next?", "What needs
to be done next?", "What comes after this?", "What do I need to do
next?", "What's the following action?", "What are the next actions
I should take?", "What action is required after this?", "What is
the subsequent step?", "What's my next move?", "What should come
next in this process?", "What's the following step?", "What's the
next task I should complete?", "What do I have to do after this?",
"What will I do next?", "What is the next thing I should do?",
"What should my next action be?", "What is my next step going to
be?", "What do I intend to do next?" ]
```

**Previous**
```
["What happened?", "What is done?", "What has been done?", "What
was done?", "What just happened?", "What did I just do?", "What
steps did I just take?", "What was the process I followed?", "What
occurred?", "What just took place?", "What actions led to this?",
"What did I just experience?", "What has just transpired?", "What
decisions did I make?", "What just went down?", "What sequence of
events led to this?", "What happened right before this?", "What
took place?", "What happened previously?", "What occurred in the
past?", "What happened back then?", "What actions were taken?",
"What happened earlier?", "What went on?", "What has occurred?" ]
```

---

## A.2 PROACTIVE ANTICIPATION ABILITY

The hyperparameters and model configurations of our trained online interaction model are summarized in Table 6. The training data originates from the training set corresponding to the benchmark usage data, constructed using the same methodology. Given the sparsity and uniqueness of special event annotations, there is no risk of data leakage.

| Parameter | Value |
|---|---|
| attn_implementation | flash_attention_2 |
| learning_rate | 0.0002 |
| optim | adamw_torch |
| lr_scheduler_type | cosine |
| lora_modules | model.*(q_proj\|k_proj\|v_proj\|o_proj\| gate_proj\|up_proj\|down_proj)\|lm_head |
| lora_r | 128 |
| lora_alpha | 256 |
| frame_fps | 4 |
| frame_resolution | 384 |
| max_num_frames | 1024 |
| frame_token_cls | True |
| frame_token_pooled | [3,3] |
| embed_mark | 4fps_384_1+3x3 |

Table 6: Training parameters of online model.

### A.3 LONG-SHORT TERM DETAILS

In the long-short-term memory mechanism, as some MLLMs generate a large number of visual tokens when encoding images/video clips, excessive redundant information accumulates in the long-term memory window. This not only consumes significant memory space but also affects the similarity calculation between adjacent clips. Therefore, in the long-term memory module, we employ average pooling to downsample the visual tokens to a specified range. The number of frame features stored in the long-term memory is set to 16. Some details are shown in Figure 7 and Algorithm 1.

---

**Prompt used for MLLM inference using long-short term memory**

```
Carefully watch the video and pay attention to the cause and sequence
of events,the detail and movement of objects, and the action and pose
of persons. Based on your observations, select the best option that
accurately addresses the question. Only give the best option.

This contains a long memory of 0.0 to {:} seconds.
<Long-term Memory>

This contains a short clip sampled from {:} to {:} seconds.
<Short-term Memory>
```

**Question:**
```
<Question>
```

---

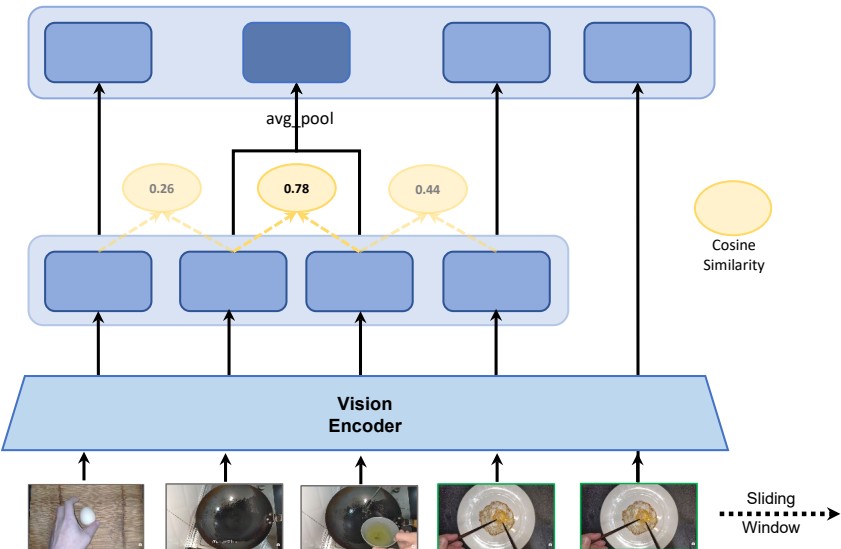

Figure 7: Design of long-term memory.

### A.4 DIFFERENT TYPES OF VISUAL CUES

To investigate model performance across different question types, we classify each question using a large language model (LLM). The classification prompt is designed to ensure balanced distribution across categories. A representative version of the prompt is as follows.

**Algorithm 1** Pseudocode of long-short term memory.

```
# video pre-processing & memory bank initialization
video = reshape_and_move_to_gpu(video)
long_embeddings = []
short_embeddings = []

# batch processing
for batch in video_batches:
   with no_grad():
      embeddings = model.encode(batch)
   long_embeddings.append(embeddings)
   short_embeddings.append(embeddings)

   # memory updating
   if len(long_embeddings) > max_memory:
      similarities = calculate_similarities(long_embeddings)
      while len(long_embeddings) > max_memory:
         merge_most_similar_pairs(long_embeddings, similarities)

# get final embeddings
final_embeddings = combine_embeddigns(long_embeddings, short_embeddings)

# get response
question = format_question(question_text, options)
response = get_model_response(model, questions, final_embeddings)
```

---

**Prompt used for classification process of different types of visual cues**

```
You are an intelligent chatbot designed to categorize video-related
questions into three predefined types based on their content.  Your
task is to analyze each question and determine which category (1, 2,
or 3) it belongs to.  Here's how you can accomplish the task:
-----------------------
```
**INSTRUCTIONS:**
```
- Carefully read the question and determine its main focus.
- Assign the question to one of the following categories:
```
- **Category 1**: Video content and details (specific scenes, objects, actions, colors, words, etc.)
- **Category 2**: Characters and events (interactions, motivations, event summaries, reasons, etc.)
- **Category 3**: Scenes and environments (locations, weather, background, video type, etc.)
```
- Your response should directly indicate the matching category
without any additional explanation or text.
- You must choose one of the three categories ('1', '2', or '3') for
each question."
Please categorize the following video-based question:
Question:  question
Provide your evaluation by directly outputting the corresponding
category ('1', '2', or '3').  DO NOT PROVIDE ANY OTHER OUTPUT TEXT
OR EXPLANATION. Only provide the matching category as a single
string.  For example, your response should look like this:  '1'."
```

## B  DATA EXAMPLE

```
{
    "video_source": "LVBench",
    "video_id": "Cm73ma6Ibcs",
    "duration_sec": 3665.5,
    "fps": 24,
```

```
        "question_id": "Cm73ma6Ibcs@1",
        "question": "What year appears in the opening caption of the
        ↪  video?",
        "choices": [
            "(A) 1636",
            "(B) 1366",
            "(C) 1363",
            "(D) 1633"
        ],
        "correct_answer": "D",
        "time_reference": [
            15,
            19
        ],
        "question_type": "Immediate",
        "question_time": 19,
        "video_path": "00000000.mp4"
    }
```

```
    {
        "video_source": "LVBench",
        "video_id": "Cm73ma6Ibcs",
        "duration_sec": 3665.5,
        "fps": 24,
        "question_id": "Cm73ma6Ibcs@1",
        "question": "What year appears in the opening caption of the
        ↪  video?",
        "choices": [
            "(A) 1636",
            "(B) 1366",
            "(C) 1363",
            "(D) 1633"
        ],
        "correct_answer": "D",
        "time_reference": [
            15,
            19
        ],
        "question_type": "Recalling@short",
        "question_time": 64.75680894210416,
        "video_path": "00000000.mp4"
    }
```

```
    {
        "video_source": "Ego-Ego4D-Narration-Val",
        "video_id": "558a082a-fcae-4cae-bcfc-5f69be53e497",
        "duration_sec": 74.5,
        "fps": 4,
        "question_id": "558a082a-fcae-4cae-bcfc-5f69be53e497@1",
        "choices": [
            "(A) You tap on the table with your fingers.",
            "(B) You push something aside.",
            "(C) You operate the mouse",
            "(D) You pull something on the table"
        ],
        "question": "When I rub the table with my hand, what should come
        ↪  next in this process?",
        "correct_answer": "A",
        "time_reference": 43.3896682,
        "question_type": "Awaiting@short",
        "question_time": 10.612453361655376,
        "video_path": "558a082a-fcae-4cae-bcfc-5f69be53e497.mp4"
    }
```

```
{
    "video_source": "Ego4D-Narration-Val",
    "video_id": "87bbc5c0-1b4a-47a5-bfbb-ec417b8e12d1",
    "duration_sec": 312.5,
    "fps": 4.0,
    "question_type": "Ongoing",
    "question_id": [
        "87bbc5c0-1b4a-47a5-bfbb-ec417b8e12d1@1",
        "87bbc5c0-1b4a-47a5-bfbb-ec417b8e12d1@2"
    ],
    "question_time": [
        270.8421305546634,
        305.95987055466344
    ],
    "question": [
        "What can you tell me about? Be concise.",
        "Simply interpret the scene for me."
    ],
    "choices": [],
    "correct_answer": {
        "87bbc5c0-1b4a-47a5-bfbb-ec417b8e12d1@1": [
            "You file a piece of furniture with a hand file.",
            "You wipe the furniture with your right hand.",
            "You remove the sellotape from the furniture."
        ],
        "87bbc5c0-1b4a-47a5-bfbb-ec417b8e12d1@2": [
            "You pick up some sellotapes from the furniture with
            ↪  your right hand.",
            "You walk to the front of the furniture.",
            "You touch the camera."
        ]
    },
    "time_reference": {
        "87bbc5c0-1b4a-47a5-bfbb-ec417b8e12d1@1": [
            270.8421305546634,
            275.80245055466344,
            277.9683505546634
        ],
        "87bbc5c0-1b4a-47a5-bfbb-ec417b8e12d1@2": [
            305.95987055466344,
            309.00634055466344,
            311.0571705546634
        ]
    }
}
```

Here are some rejected examples.

```
[
    {
        "uid": "83",
        "question": "What part of the cook is different from other
        ↪  people?",
        "answer": "D",
        "time_reference": "03:00-03:03",
        "choices": [
            "(A) Leg",
            "(B) Foot",
            "(C) Eye",
            "(D) Hand"
        ],
        "note": "easy to guess"
    },
```

```
    {
        "uid": "3790",
        "question": "What is the stoppage time for the first half?",
        "answer": "B",
        "time_reference": "47:10-47:15",
        "choices": [
            "(A) 1 minute",
            "(B) 0 minute",
            "(C) 2 minute",
            "(D) 3 minute"
        ],
        "note": "the closest one"
    },
    {
        "uid": "4167",
        "question": "What is the content of the entire video about?",
        "answer": "D",
        "time_reference": "00:00-118:28",
        "choices": [
            "(A) This video narrates the entire process of the 2016
            ↪   men's volleyball final, in which the Brazilian team
            ↪   defeated China 3-0 to win the championship",
            "(B) This video narrates the entire process of the 2016
            ↪   men's volleyball final, in which the Brazilian team
            ↪   defeated Italy 3-2 to win the championship",
            "(C) This video narrates the entire process of the 2020
            ↪   men's volleyball final, in which the Brazilian team
            ↪   defeated Italy 3-0 to win the championship",
            "(D) This video narrates the entire process of the 2016
            ↪   men's volleyball final, in which the Brazilian team
            ↪   defeated Italy 3-0 to win the championship"
        ],
        "note": "history event"
    },
    {
        "uid": "1418",
        "question": "What is the intention behind the residents showing
        ↪   the photos in the video?",
        "answer": "A",
        "time_reference": "05:13-05:59",
        "choices": [
            "(A) To inform the audience that some areas here were not
            ↪   submerged by water in the past",
            "(B) To demonstrate their photography hobby",
            "(C) To share memories of their childhood",
            "(D) To showcase their cooking skills"
        ],
        "note": "preference to longer choice"
    },
    {
        "uid": "3415",
        "question": "What might a passenger think about the vlogger from
        ↪   a third person perspective?",
        "answer": "C",
        "time_reference": "00:00-37:34",
        "choices": [
            "(A) Strange. Because he talks to his phone",
            "(B) Normal. Because he talks to himself",
            "(C) Normal. Because he talks to his phone",
            "(D) Strange. Because he talks to himself"
        ],
        "note": "easy to guess; the time reference span is too long"
    }
]
```

```
[
    {
        "video_source": "Ego-Ego4D-Narration-Train",
        "video_id": "43383e0d-02c8-4da4-8365-ccb3e6d55eac",
        "duration_sec": 100.75,
        "fps": 4,
        "question_id": "43383e0d-02c8-4da4-8365-ccb3e6d55eac@2",
        "choices": [],
        "question": "If you happen to notice 'I scroll through my phone'
        ↪  in action, I would greatly appreciate your immediate
        ↪  alert.",
        "correct_answer": "You scroll through your phone.",
        "time_reference": 98.30714719999999,
        "question_type": "Awaiting@short",
        "question_time": 67.47545180739294,
        "qa": "Person: When I scroll through my phone, what do I intend
        ↪  to do next?\nCorrect choice: Check messages\nWrong choices:
        ↪  Answer calls, Clean the floor, Turn on the lights, Put down
        ↪  the vacuum cleaner",
        "qa_question": "When I scroll through my phone, what do I intend
        ↪  to do next?",
        "qa_correct_answers": "Check messages",
        "qa_wrong_answers": [
            "Answer calls",
            "Clean the floor",
            "Turn on the lights",
            "Put down the vacuum cleaner"
        ],
        "all_choices_abc": [
            "(A) Clean the floor",
            "(B) Turn on the lights",
            "(C) Check messages",
            "(D) Answer calls",
            "(E) Put down the vacuum cleaner"
        ],
        "correct_choice_abc": "C",
        "llm_pred_qwen": "C",
        "llm_pred_llama": "C"
    },
    {
        "video_source": "Ego-Ego4D-Narration-Train",
        "video_id": "8c336ba7-c474-4d6b-ae2a-2c2388eaaca3",
        "duration_sec": 299.25,
        "fps": 4,
        "question_id": "8c336ba7-c474-4d6b-ae2a-2c2388eaaca3@1",
        "choices": [],
        "question": "Please send me an immediate alert when you notice
        ↪  'A man interacts with a woman' occurring.",
        "correct_answer": "A man interacts with a woman.",
        "time_reference": 52.741842399999996,
        "question_type": "Awaiting@short",
        "question_time": 0,
        "qa": "Person: When a man interacts with a woman, what did they
        ↪  do at that moment?\n\nCorrect choice: They started a
        ↪  conversation.\nWrong choices: \n- They sprinkled seasoning
        ↪  into the food.\n- They placed the salt container on the
        ↪  shelf.\n- They mixed French fries in a pan.\n- They adjusted
        ↪  the cookware on the cooker.",
        "qa_question": "When a man interacts with a woman, what did they
        ↪  do at that moment?",
        "qa_correct_answers": "They started a conversation.",
        "qa_wrong_answers": [
            "They sprinkled seasoning into the food.",
            "They placed the salt container on the shelf.",
```

```
                    "They mixed French fries in a pan.",
                    "They adjusted the cookware on the cooker."
                ],
                "all_choices_abc": [
                    "(A) They started a conversation.",
                    "(B) They mixed French fries in a pan.",
                    "(C) They adjusted the cookware on the cooker.",
                    "(D) They sprinkled seasoning into the food.",
                    "(E) They placed the salt container on the shelf."
                ],
                "correct_choice_abc": "A",
                "llm_pred_qwen": "A",
                "llm_pred_llama": "A"
            },
            {

                "video_source": "Ego-Ego4D-Narration-Train",
                "video_id": "9c23e7aa-226c-4596-b496-26bd9bfd03f5",
                "duration_sec": 581.5,
                "fps": 4,
                "question_id": "9c23e7aa-226c-4596-b496-26bd9bfd03f5@3",
                "choices": [],
                "question": "Should you see 'I clean a bicycle' happening,
    ↪    don\u2019t hesitate to notify me immediately.",
                "correct_answer": "You clean a bicycle.",
                "time_reference": 564.3109999999999,
                "question_type": "Awaiting@short",
                "question_time": 531.6720716807865,
                "qa": "Person: When I clean a bicycle, what happened
    ↪    before?\nCorrect choice: I took a tissue.\nWrong choices: I
    ↪    fixed the tire., I cleaned the place., I put the used tissue
    ↪    in the dustbin.",
                "qa_question": "When I clean a bicycle, what happened before?",
                "qa_correct_answers": "I took a tissue.",
                "qa_wrong_answers": [
                    "I fixed the tire.",
                    "I cleaned the place.",
                    "I put the used tissue in the dustbin."
                ],
                "all_choices_abc": [
                    "(A) I took a tissue.",
                    "(B) I put the used tissue in the dustbin.",
                    "(C) I fixed the tire.",
                    "(D) I cleaned the place."
                ],
                "correct_choice_abc": "A",
                "llm_pred_qwen": "A",
                "llm_pred_llama": "A"
            },
            {

                "video_source": "Ego-Ego4D-Narration-Train",
                "video_id": "cacb6d38-3451-4223-aa13-7ef58a6573db",
                "duration_sec": 708.25,
                "fps": 4,
                "question_id": "cacb6d38-3451-4223-aa13-7ef58a6573db@3",
                "choices": [],
                "question": "I would like to be promptly alerted if you identify
    ↪    'He washes his hands' in any situation.",
                "correct_answer": "He washes his hands.",
                "time_reference": 610.3921972,
                "question_type": "Awaiting@short",
                "question_time": 569.7305633479596,
```

```
        "qa": "Person: When he washes his hands, what's the next
        ↪  step?\nCorrect choice: He puts the soap back on the soap
        ↪  dish.\nWrong choices: \n- He puts the bowl in the sink.\n-
        ↪  He wraps the aluminum foil around the bowl.\n- He puts the
        ↪  meat in the fridge.",
        "qa_question": "When he washes his hands, what's the next
        ↪  step?",
        "qa_correct_answers": "He puts the soap back on the soap dish.",
        "qa_wrong_answers": [
            "He puts the bowl in the sink.",
            "He wraps the aluminum foil around the bowl.",
            "He puts the meat in the fridge."
        ],
        "all_choices_abc": [
            "(A) He puts the meat in the fridge.",
            "(B) He puts the soap back on the soap dish.",
            "(C) He wraps the aluminum foil around the bowl.",
            "(D) He puts the bowl in the sink."
        ],
        "correct_choice_abc": "B",
        "llm_pred_qwen": "B",
        "llm_pred_llama": "B"
    }
]
```

## C  DATASET OPEN SOURCE PROTOCOL

We adopt an **index-only release** strategy to respect original licenses and avoid video redistribution, which is shown in Table 7. We provide metadata, annotations, and download links, enabling reproducible research while ensuring legal and ethical compliance.

Table 7: Dataset licensing and our release strategy to comply with legal restrictions.

| Dataset | License / Restriction | Our Action for Release |
|---|---|---|
| LongVideoBench | CC-BY-NC-SA 4.0 (non-commercial only) | Index-only release – we provide JSON/CSV metadata and download links; videos are not re-hosted. |
| Vript-RR | Academic-use-only, no redistribution | Index-only release – we release scripts, questions, and timestamps only; users must download videos themselves under the original license terms. |
| LVBench | MIT (code) + original video copyrights | Index-only release – README explicitly states: "Video copyrights remain with original platforms; users must comply with their respective terms." |
| Ego4D | Ego4D Terms (academic, non-commercial, registration required) | Index-only release – our README directs users to the official Ego4D portal and license page for data access. No videos or clips are included. |
| QVHighlights | Original repo forbids video redistribution | Index-only release – we release only the annotations and evaluation code; README provides a direct link to the official repository for video access. |

