# OpenReview forum: "RIVER: A Real-Time Interaction Benchmark for Video LLMs"
_ICLR.cc/2026/Conference — ICLR 2026 Poster_

### Official Review · Reviewer_ya2u · 2025-10-25

**Soundness:** 2
**Presentation:** 2
**Contribution:** 2
**Rating:** 4
**Confidence:** 5

**Summary:**

RIVER Bench proposes a benchmark for real-time video interaction, shifting evaluation from offline QA to online, temporally grounded tasks. It formalizes three competencies—Retrospective Memory, Live-Perception, and Proactive Anticipation—with precise query, cue, and response timings. Built from curated long-video sources, it spans 1,067 videos and 4,278 questions, enabling forgetting-curve and latency–accuracy analyses. The authors also adapt offline MLLMs via sliding windows plus a long–short memory module and release training data that improves proactive responses. Experiments across closed- and open-source models expose gaps in online processing and show gains from RIVER-based fine-tuning.

**Strengths:**

**1. Precise task formalization and timing.** Clear definitions of Retro-Memory, Live-Perception, and Pro-Anticipation with explicit query/cue/answer timestamps yield faithful temporal grounding and enable memory-decay and response-latency analyses.

**2. Broad coverage with actionable baselines.** 1,067 videos and 4,278 questions assembled from multiple long-video sources; results reported across closed/open models with consistent prompts and a streaming setting support fairer comparisons.

**3. Practical online adaptation.** A sliding-window plus long–short memory design and a targeted training set improve proactive response metrics and repurpose offline MLLMs for streaming use.

**Weaknesses:**

**1. Novelty of method.** The “make-offline-models-online” recipe (sliding windows + long–short memory with nearest-neighbor averaging) feels incremental amid prior memory-cache approaches for streaming video.

**2. Comprehensiveness of the benchmark.** Despite 1,067/4,278 scale, sources cluster around a few public datasets (e.g., LVBench, LongVideoBench, Ego4D, QVHighlights), leaving underexplored domains like AR navigation, robotics, multi-party meetings, or broadcast sports—potentially limiting ecological validity.

**3. Audio omission.** The current release excludes audio, a core signal for live interaction (speech, alarms, ambience).

**4. Evaluator reliance and metric scope.** Open-ended answers are judged by a single LLM (Qwen2.5-72B), and Pro-Anticipation timing is reduced to response-within-window (“Res Acc”). Broader, judge-agnostic metrics (timing error, hesitation penalties, continuous narration quality) could reduce bias.

**5. Closed-source comparability.** GPT-4o/Gemini are limited to 50 frames while “online-ized” open models run at 1–4 fps; heterogeneous context budgets and sampling policies risk conflating interface limits with modeling capacity.

**6. Data generation and leakage risk.** LLM-synthesized anticipatory questions/distractors, though filtered, may import language priors; publishing templates and rejection sets would help audit artifacts and overfitting.

**7. Robustness to timestamp noise.** The benchmark assumes precise cue/query/answer times; stress-tests under timestamp jitter, dropped frames, or (future) ASR lag would better mirror deployment.

**8. Scalability and latency costs.** The long-term memory keeps 16 slots mirroring short-term token size. Growth behavior, pruning policy, and on-device latency/compute trade-offs remain under-explored.

**Questions:**

**1.** How would adding synchronized audio (speech and environmental sounds) alter forgetting curves and response timing? Which fusion (early vs. late; streaming ASR alignment) best supports Pro-Anticipation?

**2.** Can the fixed 16-slot long-term memory evolve into content-adaptive eviction/merge policies or learned key-value compression? What are the latency/recall impacts over 1-hour horizons?

**3.** Do memory and anticipation degrade differently across egocentric vs. third-person sources, scene types, or demographics? Could per-subset breakdowns and bias diagnostics guide responsible deployment?

**4.** What is agreement between different LLM judges and humans for open-ended scoring, and could pairwise preference tests or reference-free metrics capture timing quality without a single judge model?

**5.** Beyond coarse time bins, can item-response curves and psychometric difficulty by cue type (fine-grained, causal, background) support curriculum design and targeted fine-tuning?

**6.** How do results transfer to embodied/assistive agents where interruptions, safety stops, or missed cues have costs? Can RIVER simulate intervention thresholds and recovery from misfires?

**7.** Will future releases broaden domains (AR, robotics, meetings, sports) and ship licenses, prompt templates, and rejection sets so the community can audit and reliably reproduce results?

**Details Of Ethics Concerns:**

RIVER Bench’s scoring relies on a single LLM judge (Qwen2.5-72B) for open-ended answers, potentially encoding model-specific biases. The dataset aggregates from a few sources (Ego4D, LVBench, LongVideoBench, QVHighlights, Vript-RR) and uses LLM-generated/filtered questions—both risk domain and language-prior bias.

---

> ### Author Response · Authors · 2025-11-25
>
> ### **W1: Novelty of the Method**
>
> We believe that the primary contribution of this work lies not in the specific method design (sliding window + memory mechanism) but in the introducing a research framework for **real-time video interaction understanding**. Most existing video understanding benchmarks (e.g., MVBench, VideoMME, EgoSchema, MLVU) operate under an offline paradigm, using the entire video as input for general QA. While a few online benchmarks exist (VStream-QA, StreamingBench, OVBench, OVO-Bench), they largely fail to embody the interactive nature of online understanding—where the user can inquire about past, present, and future information at any moment.
>
> RIVER Bench defines the essential characteristics of real-time interaction:
> * **Streamed Input**: It mandates video input as a stream, requiring the model to acquire new frames at a fixed FPS, unlike the simplified approach of understanding static, cropped segments.
> * **Interactive Competencies**: It integrates Retrospective Memory, Live-Perception, and Proactive Anticipation, simulating an actual dynamic dialogue.
> * **Active Response**: Beyond standard QA, the model is required to autonomously determine the appropriate time to respond. RIVER Bench captures this by jointly evaluating response timing and content accuracy.
>
> In summary, RIVER Bench contributes by offering a structured platform to evaluate the necessary interactive forms and core capabilities for online video understanding, thus supporting future academic exploration and practical application in this domain.
>
> ---
>
> ### **W2, W6, Q6, Q7: Benchmark Scope, Generalizability, Open Source, and Practicality**
>
> Thank you for your valuable thoughts on video domains, generalizability, and open-source practices.
>
> **Video Scope and Generalizability (W2):**
> We confirm that Proactive Anticipation data primarily uses Ego4D-Narration (first-person) and QVHighlights (third-person). Ego-Narration focuses on simpler, daily, static tasks, while QVHighlights, along with the sources for Retrospective Memory (Vript-RR, LVBench, LongVideoBench), covers a broad range of general domains: travel, food, education, sports, news, and daily life. While a general video benchmark should be broad, high-precision domains like traffic flow prediction or industrial fault detection, with their unique video styles and stringent accuracy requirements, are best assessed using specialized, dedicated benchmarks to ensure accuracy and stability. We are committed to inspecting and expanding the general video domains in the future to ensure RIVER Bench has robust generalizability.
>
> **Embodied Agents and Practicality (Q6):**
> For embodied or assistive agent scenarios, the requirement for instantaneous response is extremely high, as misjudgments can lead to severe harm or loss. We therefore recommend using specialized, customized datasets for these critical applications, rather than a general video interaction dataset intended for evaluating comprehensive model capabilities. Consistent with our discussion for Reviewer HLZ6-Q1, the model's ability to output confidence/uncertainty is vital in real-world settings—it allows the agent to assess ambiguous events and potentially recover from a false trigger by restarting detection.
>
> **Open Source and Auditability (W6, Q7):**
> To facilitate community auditing and reliable reproduction of results, we commit to a comprehensive release in the revised paper, including:
>
> * Public Licenses (and a compliance strategy for restricted sources).
> * Detailed Prompt Templates used for evaluation and data generation.
> * Rejection Sets (filtered samples) to allow auditing of potential artifacts.
> * Open-source dataset and evaluation scripts.

---

> ### Author Response · Authors · 2025-11-25
>
> ### **W3 & Q1: Omission of ASR**
>
> We agree that audio data is a crucial signal in real-time interaction. To supplement the experiments in this area, we collected the audio data corresponding to the videos and used the **Whisper** model to extract subtitles, which were input to the model as text for evaluation.The results in Table 1 show the impact of ASR text on the accuracy of the **Retrospective Memory** task.
>
> **Table 1: Comparison of Retrospective Memory Accuracy (%) with and without ASR (Whisper Subtitles).**
> | Model | Short | Middle | Long | Very Long | Avg |
> | :--- | :---: | :---: | :---: | :---: | :---: |
> | **w/o ASR** | | | | | |
> | VideoChat2-HD | 34.25 | 36.19 | 33.43 | 32.60 | 34.19 |
> | InternVL2.5 | 37.02 | 39.50 | 32.60 | 30.66 | 36.33 |
> | LLava-Video | 44.20 | 43.65 | 37.02 | 37.29 | 40.68 |
> | VideoChat-Flash | 43.92 | 48.90 | 41.44 | 38.12 | 44.34 |
> | **w/ ASR** | | | | | |
> | VideoChat2-HD | 39.50 | 37.85 | 37.57 | 32.04 | 36.74 |
> | InternVL2.5 | 45.86 | 46.96 | 39.50 | 34.55 | 41.72 |
> | LLava-Video | 47.51 | 44.48 | 44.20 | 35.87 | 43.01 |
> | VideoChat-Flash | 50.55 | 52.49 | 46.13 | 39.23 | 47.10 |
>
> ---
> ### **W4-1 & Q4: Metric of Proactive Anticipation**
>
> First, regarding the issue of **LLM bias**, we addressed it by using **multiple LLMs** to score the responses and comparing the results. The consistency across different LLMs demonstrates the robustness of our evaluation and ensures that the scoring does not overly depend on a single model. Additionally, using open-source models prevents issues such as **weight changes** or **API failures**. We further optimized the scoring process by using **yes/no checklist queries**, which are generated based on the video annotations. This fine-grained breakdown provides more accurate scores by indicating how many queries the predicted outputs successfully recalled. This simplified question format makes it easier to provide fair evaluations that are not influenced by any specific LLM's biases. In the future, we plan to release the checklist queries and prompts used in the evaluation process to ensure transparency and reproducibility.
>
> We also conducted **ablation studies** on the **prompting** strategies, testing different levels of strictness. The results showed good consistency across these prompt variations, which adds confidence to the robustness of our evaluation procedure.
>
> **Table 2: Ablation experiments on metrics of Proactive Anticipation across different models.**
>
> | Judge / Prompt                | VideoChat2-HD | InternVL2.5 | LLaVA-Video | VideoChat-Flash |
> |--------------------------------|---------------|-------------|-------------|-----------------|
> | Qwen2.5-72B (original)         | 4.25          | 4.09        | 5.72        | 5.64            |
> | GPT-4o                         | 3.89          | 3.74        | 5.51        | 5.32            |
> | Qwen2.5-72B + checklist query  | 3.30          | 3.26        | 5.86        | 5.66            |
> | GPT-4o + checklist query       | 3.21          | 3.13        | 5.74        | 5.57            |
> | Qwen2.5-72B strict             | 0.74          | 0.74        | 1.72        | 1.31            |
> | Qwen2.5-72B loose              | 7.20          | 7.52        | 11.45       | 10.38           |

---

> ### Author Response · Authors · 2025-11-25
>
> ### **W4-2: Expanded Evaluation Metrics (Timing Error)**
>
> Thank you for your suggestion. We have expanded the evaluation metrics for the **Proactive Anticipation** task to more comprehensively assess temporal accuracy and user experience. The results, showing strong consistency, are presented in Table 3 below.
>
> We introduced two metrics for temporal precision:
> * **Abs Time Error (Absolute Time Error):** The mean absolute error (in seconds) between the model's response time and the event's start time.
> * **Time Error Score:** A piecewise-designed metric intended to better simulate human subjective perception of the Proactive Anticipation task: tolerance for acceptable timing errors, dissatisfaction with early responses (false triggers), and reluctant acceptance of late responses.
>
> **Table 3: Expanded Temporal Accuracy Metrics for the Proactive Anticipation Task.**
> | Model | Abs Time Error (s) | Time Error Score |
> | :--- | :---: | :---: |
> | VideoChat2-HD | 171.5 | 4.51 |
> | InternVL2.5 | 134.3 | 17.95 |
> | LLava-Video-Qwen | 65.5 | 19.50 |
> | VideoChat-Flash | 40.8 | 20.24 |
>
> ---
> ### **W5 & Q1: Computational Resources and Fairness**
>
> Thank you for your attention to computational resources and budget fairness. We prioritized equitable comparison in designing our evaluation protocol, implementing the following specific measures:
>
> 1.  **Closed-Source Model Budget Clarification:**
>     For closed-source models, we uploaded video files and tested them via black-box API calls, which restricts flexible internal design adjustments. We clarify the frame definitions: the standard configuration for **GPT-4o is 64 frames** per query, and **Gemini 1.5 Pro operates at 1 FPS** for streaming input.
>
> 2.  **Open-Source Input and Token Alignment:**
>     For open-source video understanding models, we maintained consistency with their official code's basic input configuration. For the memory module (Long-Term Memory, LTM), we ensured consistency in the number of memory slots ($M$) and the total token count. Specifically for VideoChat2-HD, which uses a Q-Former structure making direct adjustment difficult, we conducted supplementary experiments: we expanded its memory slots by a factor of **1.5** (from $M$ to $1.5M$) to precisely align the total LTM token budget across all models.
>
> **Table 4: Details on Token Alignment for Short-Term ($F$ frames) and Long-Term ($M$ slots) Memory.**
> | Model | Post-ViT Feature Size | Connector | Short-Term (ST) Tokens | Long-Term (LTM) Tokens |
> | :--- | :--- | :--- | :--- | :--- |
> | VideoChat2-HD | [$F$, 196, 1024] | Q-Former | [96, 4096] | [$M \times 96, 4096$] $\rightarrow$ [**$1.5M \times 96, 4096$**] |
> | InternVL2.5 | [$F$, 256, 1024] | MLP | [$F \times 256, 4096$] | [$M \times 144, 4096$] |
> | LLava-Video | [$F$, 196, 1024] | MLP | [$F \times 196, 3584$] | [$M \times 144, 3584$] |
> | VideoChat-Flash | [$F$/4, 784, 1024] | ToMe + MLP | [$F/4 \times 64, 3584$] | [$M \times 144, 3584$] |
>
>
> 3.  **Inference Latency Reporting:**
>     Addressing the strict latency requirements for real-time interaction, we measured the average inference latency (in seconds) for each model, covering three key stages: image encoding, decision/connector processing, and average response generation.
>
> **Table 5: Average Latency (seconds) for Models Across Different Inference Stages.**
> | Model | Image Encoding (s) | Decision/Connector (s) | Avg Response Latency (s) |
> | :--- | :---: | :---: | :---: |
> | VideoChat2-HD | 0.365 | 0.162 | 0.635 |
> | InternVL2.5 | 0.077 | 0.578 | 0.847 |
> | LLava-Video | 0.174 | 0.433 | 0.297 |
> | VideoChat-Flash | 0.021 | 0.096 | 0.557 |

---

> ### Author Response · Authors · 2025-11-25
>
> ### **W7: Robustness to Timestamp Noise**
>
> You raise a very critical point. The current benchmark indeed operates under idealized timestamp assumptions, which facilitates controlled variable testing in lab evaluations. However, we acknowledge that this differs from **real-world scenarios** that involve ASR latency, frame synchronization errors, or temporal jitter. We fully agree that future work must introduce such perturbations (e.g., controlled time offsets, frame drop simulation, or audio-video alignment noise) to build a **robust evaluation protocol**. This will more genuinely reflect the model's performance in the actual deployment environment. This work is foundational for real-time video interaction evaluation, and future extensions will focus on practical scenario adaptation.
>
> ---
> ### **W8 & Q2: Scalability and Memory Extensibility**
>
> Thank you for your suggestion. Our current memory module utilizes a non-parametric approach, which ensures it is **lightweight, highly transferable, and extensible**. For edge device deployment, users can select different memory slots, compression ratios, and context lengths based on device performance to meet the specific **latency-performance trade-off** (some ablation results are referenced in the L8cs-Q1 response). For*extremely long videos (exceeding 3600s), which constitute 68.23% of the Retrospective Memory task samples, our memory method can maintain good recall capability, as supported by our experimental data.
>
> ---
> ### **Q3: Distribution Differences and Bias Diagnostics**
>
> Currently, our experiments focus on datasets with mixed perspectives and general scenarios. This detailed analysis is vital for discovering potential biases and preventing model failure in specific groups or scenarios. We agree that future work should employ **fine-grained** evaluation and fairness diagnostic tools (e.g., performance comparison by group, calibration error analysis) to identify systemic bias and establish **more responsible deployment strategies**—for instance, setting performance thresholds for high-risk subsets. Particularly in **high-risk or high-precision scenarios** like traffic control, industrial inspection, and smart companionship, model reliability directly impacts safety and user experience, necessitating specialized memory architecture design and robust validation coupled with domain knowledge.
>
> ---
> ### **Q5: Differentiating Cognitive Difficulty by Cue Type**
>
> Thank you for this suggestion. We categorized questions into **fine-grained, causal, and background** types and observed clear performance disparities (e.g., **causal** questions were significantly more difficult). Your proposal to use this to build Item-Response Curves and psychometric difficulty metrics is valuable for informing curriculum design and targeted fine-tuning. In future work, we will introduce methods like Item-Response Theory (IRT) to systematically quantify the cognitive difficulty of different question types and design more effective training strategies. **Causal reasoning**, in particular, could be explored as a dedicated research area, as significant advancements here could substantially enhance the inference capabilities, reasoning power, and overall intelligence of current multimodal models.

---

> > ### Comment · Reviewer_ya2u · 2025-11-26
> >
> > Thanks for your response. My concerns have partially been resolved.

---

> ### Author Response · Authors · 2025-11-26
>
> Thank you very much for your prompt reply.
>
> Our previous response regarding W1 was insufficiently specific and precise, so we have revised it. Our contribution can be summarized in three main parts:
>
> - **Streamed Input & Real-time Interactive Tasks**: Our starting point was to establish a task format for **real-time interaction**, building a relatively comprehensive set of interactive tasks. The model captures continuous video stream input and flexibly and proactively responds to human inquiries.
> - **Evaluation Metric**: We then proposed several evaluation metrics to quantify the model's real-time interaction capabilities, simultaneously reflecting both the **accuracy** and the **timeliness** of the response.
> - **Active Response**: Finally, we provided a general and easily transferable method for **converting** existing offline video understanding models into online understanding models, thereby granting the models an active response capability. These methods, on one hand, offer a **baseline** for online interaction that future research can reference, and on the other hand, they allow us to **validate** the reasonableness of our quantitative evaluation metrics.
>
> The extensive experimental data presented in the main paper and during the rebuttal phase demonstrates that our proposed **quantitative metrics** are sound and that the **conversion** of offline models into online models is effective, which facilitates community reference and further research.
>
> Please feel free to raise any further questions you may have. We are very willing to continue the dialogue.

---

### Official Review · Reviewer_HLZ6 · 2025-10-31

**Soundness:** 2
**Presentation:** 3
**Contribution:** 2
**Rating:** 4
**Confidence:** 3

**Summary:**

This paper presents RIVER Bench (Real-tIme Video intERaction Bench), a new benchmark designed to evaluate online and real-time video understanding in multimodal large language models (MLLMs). Unlike traditional offline settings, RIVER Bench defines three interactive task types — Retrospective Memory, Live Perception, and Proactive Anticipation — to simulate dynamic, conversational interactions with ongoing video streams. Comprehensive annotations and evaluations reveal that while existing offline MLLMs perform well on static QA tasks, they struggle with real-time comprehension and long-term consistency. To address this, the authors propose a general enhancement method that improves model adaptability and responsiveness in real-time scenarios, setting a foundation for future interactive video models.

**Strengths:**

1. The paper fills a clear gap by introducing RIVER Bench for real-time video interaction, moving beyond the traditional offline paradigm. Its design with Retrospective Memory, Live Perception, and Proactive Anticipation tasks realistically mimics dynamic, interactive scenarios.

2. The dataset is diverse and well-annotated, combining multiple video sources with fine-grained temporal labeling and strong quality control, ensuring high reliability.

3. The experiments are comprehensive, covering various model types and introducing well-designed metrics (e.g., response localization, memory decay) that align with real-time understanding needs.

4. The method is simple yet effective, using a sliding-window and long-short memory mechanism to adapt offline models for online use, showing clear performance gains (+11.28% on RIVER).

**Weaknesses:**

1. RIVER Bench only supports video-text interaction, not including audio, which is crucial for real-time tasks like voice navigation or human-robot interaction. While this is mentioned as a limitation, it would be helpful to test and report ASR performance, which would increase the benchmark’s practical value.

2. The data primarily comes from Ego4D-Narration, focusing on simple, static tasks like desk operations and furniture organization, and lacks more complex dynamic scenarios such as traffic flow prediction or industrial fault detection. This could limit the benchmark’s generalizability to real-world applications.

3. Many experimental settings lack ablation studies, such as the impact of memory compression strategies or the number of memory slots. Additionally, inference latency (e.g., per-frame processing time, response delay) is not reported, which is crucial for real-time interaction. Key details like frame sampling rates and the computational cost of memory modules are missing, making it hard to assess the feasibility of this method in actual deployments.

**Questions:**

1. How is the response time window for the Proactive Anticipation task defined (e.g., fixed ±1s, or dynamically adjusted based on event duration)? Will the authors consider adding an uncertainty prediction metric (e.g., model confidence in its predictions) to assess decision-making in ambiguous scenarios?

2. What is the proportion of samples in the Retrospective Memory task for the "extremely long duration" (>3600s) category?

3. The experiments show that all models perform poorly on causal clue (CC) questions. What specific technical directions do the authors suggest for improving the integration of visual perception and event attribution in such tasks?

---

> ### Author Response · Authors · 2025-11-25
>
> ### **W1: Adding ASR Performance**
>
> Thank you for your valuable feedback regarding the inclusion of ASR performance. In response, we have incorporated the audio component by collecting audio data corresponding to the video and using Whisper for speech-to-text conversion. The text generated by ASR is then input into the model.
>
> We observed that the inclusion of ASR led to noticeable improvements in the **retroactive memory** task. This improvement can likely be attributed to the fact that subtitles contain more prompting information* which enhances the model's ability to recall relevant events. However, for **Live Perception**, there was a slight decline in performance. This could be due to the fact that the tasks focused more on real-time video frames, and the real-world dialogue or subtitles often suffer from delays, unlike live or instructional videos, which provide real-time commentary. Inaccurate or mismatched ASR output may have caused interference in the model’s real-time perception capabilities.
>
> **Table 1: ASR Performance Comparison**
>
> | Model            | | | | |  Retro-Memory (Acc) | Live-Perception (Acc) | Pro-Anticipation (Loc) |
> |--------------------|-------|---|---|---|---------|----------------------|------------------------|
> | | **Short**   | **Middle**   | **Long** | **Very Long** | **Avg** | **Acc** | **Loc** |
> | **w/o ASR** |            |          |                       |               |           |         |
> | VideoChat2-HD    | 34.25             | 36.19                | 33.43                  | 32.60         | 34.19     | 41.16   | 2.16    |
> | InternVL2.5      | 37.02             | 39.50                | 32.60                  | 30.66         | 36.33     | 58.84   | 16.65   |
> | LLava-Video      | 44.20             | 43.65                | 37.02                  | 37.29         | 40.68     | 51.38   | 22.51   |
> | VideoChat-Flash  | 43.92             | 48.90                | 41.44                  | 38.12         | 44.34     | 56.35   | 20.45   |
> | **w/ ASR**       |                   |                      |                        |               |           |         |
> | VideoChat2-HD    | 39.50             | 37.85                | 37.57                  | 32.04         | 36.74     | 34.81   | 4.54    |
> | InternVL2.5      | 45.86             | 46.96                | 39.50                  | 34.55         | 41.72     | 45.86   | 20.15   |
> | LLava-Video      | 47.51             | 44.48                | 44.20                  | 35.87         | 43.01     | 44.20   | 18.15   |
> | VideoChat-Flash  | 50.55             | 52.49                | 46.13                  | 39.23         | 47.10     | 45.03   | 21.05   |
>
> ---
> ### **W2: Scene Diversity in the Dataset**
>
> We acknowledge your concern regarding the domain focus and generalizability of the benchmark data. We confirm that during data construction, the **Proactive Anticipation** task primarily utilizes items from **Ego4D-Narration** (first-person perspective) and **QVHighlights** (third-person perspective).
>
> * As you rightly noted, **Ego4D-Narration** focuses on simpler, more static tasks like desk operations and furniture organization.
> * **QVHighlights**, conversely, encompasses more diverse topics such as daily activities, travel experiences, news and current affairs, sports, cooking, education, and entertainment.
>
> Furthermore, the **Retrospective Memory** task integrates sources from **Vript-RR, LVBench, and LongVideoBench**, covering a broad spectrum of domains including travel, food, daily life, instructional videos, documentaries, education, sports, gaming, live stream replays, and news.
>
> We agree that a universal video benchmark should encompass a wide range of fields to ensure strong generalizability. However, highly specialized fields, such as traffic flow prediction or industrial fault detection, which require extreme precision and feature unique video styles, are often better evaluated using **dedicated benchmarks** to ensure accuracy and stability.
>
> We will review and actively expand the video domains in our future work to ensure that RIVER Bench, as a general video interaction benchmark, possesses robust generalizability.

---

> ### Author Response · Authors · 2025-11-25
>
> ### **Memory Module**
>
> #### **Memory Slots**
> For the **memory slots** ablation, we used **VideoChat2** as the model of choice. As the number of memory slots increased, we observed a consistent improvement in model performance.
>
>
> **Table 4: Performance of VideoChat2 with Varying Memory Slots**
>
> | Slots Num | Short | Middle | Long | Very Long | Avg |
> |-----------|-------|--------|------|-----------|-----|
> | 8         | 34.53 | 35.36  | 32.32| 30.39     | 33.15|
> | 16        | 34.53 | 36.46  | 33.15| 32.60     | 34.19|
> | 24        | 34.25 | 37.29  | 34.53| 32.87     | 34.74|
> | 32        | 35.91 | 37.02  | 34.53| 31.77     | 34.81|
> | 48        | 35.08 | 37.85  | 35.36| 33.15     | 35.36|
>
> ---
>
> #### **Compression Ratio**
> For the **compression ratio**, we conducted experiments using the **LLava-Video** model, which employs a **ViT-connector-LLM** structure, making it easier to assess the effect of compression. As the compression size increased, model performance continued to improve.
>
> **Table 5: Performance of LLava-Video with Varying Compressed Slot Sizes**
>
> | Compressed Size of Each Slot | Short | Middle | Long | Very Long | Avg |
> |------------------------------|-------|--------|------|-----------|-----|
> | 3                            | 44.20 | 43.65  | 37.02| 37.85     | 40.68|
> | 4                            | 48.07 | 45.86  | 40.51| 37.57     | 43.00|
> | 5                            | 48.90 | 48.34  | 43.65| 38.40     | 44.82|
> | 6                            | 48.43 | 50.79  | 43.22| 38.80     | 45.29|
> | 9                            | 48.62 | 50.28  | 42.54| 40.05     | 45.37|
>
> ---
> #### **Computation Cost**
> For closed-source models, we tested them via black-box API calls by uploading video files, which inherently restricts our ability to flexibly adjust their internal design. We clarify the frame rate/budget definition for these models: the standard configuration for GPT-4o is 64 frames per query, and for Gemini 1.5 Pro it is 1 FPS. For open-source video understanding models, we maintained consistency with their official code's basic input configuration, regardless of their specific image encoder or connector structure.
>
> For the long-term memory (LTM) module, we strived to ensure that both the number of memory slots ($M$) and the total LTM token count were aligned across adapted models for a fair comparison. Since the Q-Former structure in VideoChat2-HD makes direct token count alignment challenging, we conducted an additional experiment: we expanded the number of memory slots for VideoChat2-HD by a factor of 1.5 (from $M$ to $1.5M$) to precisely match its total LTM token count with the other models. The specific details on token counts are summarized in the table below (further ablations are provided in our response to Q1).
>
> **Table6: Details on Token Alignment for Open-Source MLLMs. *F* represents the number of video frames used for the Short-Term Memory input window. *M* represents the number of Long-Term Memory slots used by the model.**
> | | Visual Feature Size | Connector Type | Short-Term Token Count | Long-Term Token Count |
> | :--- | :--- | :--- | :--- | :--- |
> | VideoChat2-HD | [$F$, 196, 1024] | Q-Former | [96, 4096] | [$M \times 96, 4096$] $\rightarrow$ [**$1.5M \times 96, 4096$**] |
> | InternVL2.5 | [$F$, 256, 1024] | MLP | [$F \times 256, 4096$] | [$M \times 144, 4096$] |
> | LLava-Video | [$F$, 196, 1024] | MLP | [$F \times 196, 3584$] | [$M \times 144, 3584$] |
> | VideoChat-Flash | [$F$/4, 784, 1024] | ToMe + MLP | [$F$/4 $\times 64, 3584$] | [$M \times 144, 3584$] |

---

> ### Author Response · Authors · 2025-11-25
>
> ### **Q1: Awaiting Window Size and Uncertainty Prediction**
>
> Thank you for your thoughtful comments. To enable general offline models to handle real-time responsiveness, we employed a sliding window approach to capture video segment inputs. For consistency in evaluation, the time window for **Proactive Anticipation** matches the model’s sliding window size. More details on the window size ablation can be found in **Reviewer 9Ctw’s W1-2** response.
>
> Regarding **uncertainty prediction**, we agree that this is a critical aspect, especially in ambiguous scenarios where defining the start and end of events can be challenging. In the paper, we employed a training-free approach where the model is prompted with a predefined prompt and outputs a prediction score. However, to be honest, this method does not fairly output confidence levels because the prediction scores are influenced by the **LLM’s own confidence**. As seen in **Reviewer L8cs’s Table 2** for **trigger-related data**, overly confident models tend to output high confidence scores blindly, while less confident models may be more cautious. However, after event localization, the models that were previously overconfident do not necessarily have the advantage during follow-up inquiries. This brings up the interesting issue of "whether the model knows what it doesn't know." Overall, I would recommend using a small external trainable decision model to predict the confidence levels more fairly.
>
> ---
>
> ### **Q2: Data Proportion**
>
> In the **Retrospective Memory** task, the "extremely long videos" (videos longer than 3600 seconds) accounts for **68.23%** of the data. Given that most videos need to accommodate a variety of question types, including short (15-30s), middle (30-60s), long (300-900s), and very long (1800-3600s), we used a large number of longer videos to ensure that the benchmark can evaluate the model’s ability to handle videos of varying lengths.
>
> ---
>
> ### **Q3: Method for Causal Clue Questions**
>
> Currently, visual-language models generally perform poorly on answering causal clue questions. This is primarily because these models still struggle to connect visual events with the underlying causes. Many times, the models rely on temporal order or linguistic common sense to “guess” the answer, rather than retracing the video and identifying the actual causal basis. To improve this, we believe progress needs to be made in a few key directions:
>
> 1. **Explicit Causal Modeling**: Incorporating explicit causal relationship modeling in the model architecture could enhance the ability to connect visual cues with causal events.
> 2. **Better Long-term History Retrieval**: Improving the model’s ability to retrieve and associate long-term historical video segments would help it better understand causal links.
> 3. **Training with Causal Reasoning Tasks**: Incorporating tasks that require **causal reasoning**, such as counterfactual or intervention questions, during training could better prepare the model for causal inference tasks.
> 4. **Causal Chain Datasets**: Constructing clearer causal chain annotated datasets would also help the model improve its causal reasoning ability.
>
> We believe that these steps would provide significant improvements in the model’s performance on causal clue questions, and we plan to explore them further in future work.
>
> ---
>
> We appreciate your insightful suggestions, and we will incorporate these improvements and clarifications in the revised version of the paper.

---

### Official Review · Reviewer_9Ctw · 2025-11-01

**Soundness:** 3
**Presentation:** 3
**Contribution:** 3
**Rating:** 6
**Confidence:** 3

**Summary:**

This paper proposes RIVER Bench, a benchmark for online video interaction evaluating multimodal LLMs (MLLMs) on three temporally grounded competencies: live-perception, retrospective memory, and proactive anticipation. The authors formalize an online video-text-to-text task with explicit timestamps for query, clue, and response. They offer task taxonomies, dataset statistics, and metrics that jointly assess accuracy, timeliness, and latency–accuracy tradeoffs. RIVER aggregates and restructures items from Vript-RR, LongVideoBench, LVBench, Ego4D, and QVHighlights into precisely timed QA pairs and streaming tasks. The evaluations make a comparison between public and closed-sourced MLLMs, and demonstrate sizable gaps between offline QA success and real-time performance.

**Strengths:**

1. The paper creates an accurate online task formalization with retro/live/pro-anticipation split and timing semantics. In addition, the windowed formulation ties the accuracy to when the answer is generated.
2. Curated and broad construction across long videos, equipped with explicit filtering to mitigate language-only priors and ambiguous items.
3. Operational online protocol that makes many offline models evaluable in real time, which enables informative cross-family comparisons.
4. Comparatively useful metrics and analyses, which include duration-stratified forgetting curves, cue-type breakdowns and proactive Res Acc.

**Weaknesses:**

1. OE scoring depends on Qwen2.5-72B; prompts, thresholds, and sensitivity analyses are not deeply reported. The Res Acc window width and tolerance are not exhaustively justified; in addition, user-centric latency–utility tradeoffs are not evaluated as well.
2. From my perspective, even though the pipeline filters items that are language-answerable, deep LLM participation risks unintended stylistic mimicry and inherent bias.. More transparent human IAA and QA metrics would be quite helpful.
3. The non-public models use the determined 50 frames the meanwhile, others utilize 16 frames or 1 frame per second streams; the differences in budgets are able to confound absolute rankings. More apples-to-apples ablations would strengthen the claims to a great extent.

**Questions:**

1. Can you add a study equating tokens/frames/FLOPs across models (e.g., standardizing to a fixed #frames or fps) to isolate modeling differences from budget differences?

---

> ### Author Response · Authors · 2025-11-25
>
> ### **W1-1: Open-Ended Questioning**
>
> Thank you for your insightful question. We conducted **ablation experiments** on both the **prompting strategy** and the selection of **LLMs**, and the results consistently showed that the evaluation was **fair and unbiased** across different models. This addresses the concern regarding the influence of LLM biases on the scoring. Additionally, your concern is quite similar to **Reviewer L8cs's W2, W3-2, Q2-1**, and if you're interested, you can refer to the detailed response I provided to them for further clarification on this matter.
>
> **Table 1: Performance of Different Models Under Various Prompt Conditions**
>
> | Type            | VideoChat2-HD | InternVL2.5 | LLava-Video | VideoChat-Flash |
> |-----------------|---------------|-------------|-------------|-----------------|
> | **Standard**    | 4.58          | 4.09        | 6.21        | 6.21            |
> | **Strict**      | 0.74          | 0.71        | 1.72        | 1.31            |
> | **Loose**       | 7.20          | 7.52        | 11.45       | 10.38           |
> | **Standard (GPT-4o)** | 3.89    | 3.94        | 5.51        | 5.32            |
>
> The table presents the evaluation scores for different **prompt types** (standard, strict, and loose) across multiple models. These results reinforce that the scoring is consistent and not biased by any particular prompting or LLM choice. We hope this clears up any concerns regarding the fairness of the evaluation process.
>
> ### **W1-2: Awaiting Window Width and User Experience**
>
> Thank you for your comment. Similar to **Reviewer L8cs's W3-1 and Q2-2**, we focused on exploring the **window width** ablation and its impact on **user experience**. Our experiments show that the most significant difference in scores occurs between a **0.5 window** and a **1.0 window**. Expanding the window further does not continue to improve scores; instead, it broadens the evaluation too much, making the results less meaningful.
>
> In addition, we expanded the **time error** metric, which includes:
>
> - **Absolute Time Error (ATE)**: The average absolute error between the response time and the event start time.
> - **Time Error Score**: This is a segmented design that reflects **human subjective tolerance** for timing errors in the **Proactive Anticipation** task. The score accounts for acceptable timing error ranges, penalizes early responses, and reluctantly accepts delayed responses.
>
> For more details on these extended metrics, you can refer to the **reviewer L8cs's W3-1 and Q2-2** for similar concerns and findings.
>
> **Table 2: Performance Metrics for Varying Window Sizes**
>  | Model | Abs Time Error (s) | Time Error Score | Res Acc (0.5 window) | Res Acc (1.0 window) | Res Acc (1.5 window) | Res Acc (2.0 window) |
>  | :--- | :---: | :---: | :---: | :---: | :---: | :---: |
>  | VideoChat2-HD | 171.5 | 4.51 | 1.47 | 2.16 | 2.16 | 2.16 |
>  | InternVL2.5 | 134.3 | 17.95 | 12.15 | 16.65 | 16.76 | 16.86 |
>  | LLava-Video | 65.5 | 19.50 | 17.30 | 22.51 | 23.82 | 23.94 |
>  | VideoChat-Flash | 40.8 | 20.24 | 16.45 | 20.45 | 20.74 | 21.05 |
>
> ---
> ### **W2: Consistency and Bias Mitigation**
>
> Thank you for your question. Our goal was to implement filtering processes to avoid biases or preferences that may arise from the **LLMs** used by the **VLMs** (Video-Language Models). We sought to prevent LLMs from easily inferring the correct answer based on language-only priors. Despite our best efforts, we acknowledge that some stylistic influences or biases may still be introduced during the filtering and question generation process.
>
> To address this, we first modified the generation prompts and templates multiple times to ensure that the generated questions remained aligned with the video content and did not deviate significantly. This process aimed to prevent the generation of low-quality questions or options that were overly biased or irrelevant to the video content. After this, the data underwent manual verification, and about 90% of the questions were retained.
>
> For a more intuitive understanding of this process, we refer you to **Appendix B** in our paper, where we provide a demo of the question generation and filtering procedure. In the revised version of the paper, we will include additional details on the intermediate steps and provide more examples of **sampled** and **unsampled** questions to clarify the process further.

---

> ### Author Response · Authors · 2025-11-25
>
> ### **W3 & Q1: Computational Resources**
>
> Your concern about fairness regarding model sampling budget and the implementation details of the memory module is highly pertinent, and these factors were primary considerations in our design.
>
> For closed-source models, we tested them via black-box API calls by uploading video files, which inherently restricts our ability to flexibly adjust their internal design. We clarify the frame rate/budget definition for these models: the standard configuration for GPT-4o is 64 frames per query, and for Gemini 1.5 Pro it is 1 FPS. For open-source video understanding models, we maintained consistency with their official code's basic input configuration, regardless of their specific image encoder or connector structure.
>
> For the long-term memory (LTM) module, we strived to ensure that both the number of memory slots ($M$) and the total LTM token count were aligned across adapted models for a fair comparison. Since the Q-Former structure in VideoChat2-HD makes direct token count alignment challenging, we conducted an additional experiment: we expanded the number of memory slots for VideoChat2-HD by a factor of 1.5 (from $M$ to $1.5M$) to precisely match its total LTM token count with the other models. The specific details on token counts are summarized in the table below (further ablations are provided in our response to Q1).
>
> **Table3: Details on Token Alignment for Open-Source MLLMs. *F* represents the number of video frames used for the Short-Term Memory input window. *M* represents the number of Long-Term Memory slots used by the model.**
> | | Visual Feature Size | Connector Type | Short-Term Token Count | Long-Term Token Count |
> | :--- | :--- | :--- | :--- | :--- |
> | VideoChat2-HD | [$F$, 196, 1024] | Q-Former | [96, 4096] | [$M \times 96, 4096$] $\rightarrow$ [**$1.5M \times 96, 4096$**] |
> | InternVL2.5 | [$F$, 256, 1024] | MLP | [$F \times 256, 4096$] | [$M \times 144, 4096$] |
> | LLava-Video | [$F$, 196, 1024] | MLP | [$F \times 196, 3584$] | [$M \times 144, 3584$] |
> | VideoChat-Flash | [$F$/4, 784, 1024] | ToMe + MLP | [$F$/4 $\times 64, 3584$] | [$M \times 144, 3584$] |

---

### Official Review · Reviewer_L8cs · 2025-11-11

**Soundness:** 3
**Presentation:** 3
**Contribution:** 3
**Rating:** 6
**Confidence:** 2

**Summary:**

The paper targets online video interaction for MLLMs and proposes RIVER Bench, a benchmark and protocol to evaluate three competencies under streaming conditions: Retrospective Memory (recall past events; forgetting curves), Live-Perception (time-sensitive understanding with latency–accuracy trade-offs), and Proactive Anticipation (detect/forecast future states with correct timing). RIVER reformulates and precisely timestamps items from multiple sources (Vript-RR, LVBench, LongVideoBench, Ego4D, QVHighlights), yielding: Retro-Memory (≈1.5k MCQs across four recall intervals), Live-Perception (≈0.4k items), Pro-Anticipation (≈1.2k “stream” continuous narration + ≈1.4k “instant” trigger items). Quality control uses LLM + human filtering and distinctive time-anchored events. Experiments cover commercial models, “native online” models, offline models adapted via a sliding window + long/short-term memory wrapper (16 slots with NN-averaged compression), and a fine-tuned online model (SiGLIP encoder + LLaMA3-8B with LoRA; LM + streaming losses). Metrics include MC exact match, open-ended judging via Qwen2.5-72B, and a response accuracy (Res Acc) that credits answers emitted within a ground-truth time window. Findings: offline models that excel at single-turn QA struggle online; the proposed online adaptation narrows the gap and improves pro-anticipation (+11.28% over baseline); memory modules flatten forgetting by ~12%; causal-cue retro-memory is hardest.

**Strengths:**

•	Clear, timely problem: shifts evaluation from offline video QA to interactive, streaming settings with explicit timing of query/cue/response.

•	Three-facet design: jointly measures recall, live perception, and anticipation, and ties performance to the temporal gap \Delta (forgetting/anticipation curves).

•	Protocol precision: items carry exact timestamps; “instant” vs “stream” anticipation reflects real interaction patterns (trigger vs continuous narration).

•	Methodological baselines: shows how to wrap offline models for online inference (sliding window + memory), a practical recipe many will try.

•	Analyses: memory-duration breakdowns; cue-type breakdown (fine-grained / causal / background); evidence that more frames/time help under this protocol (unlike some offline MCQ benches).

**Weaknesses:**

•	Data novelty & provenance: benchmark reuses existing datasets; novelty is primarily the reconstruction into an online protocol. That’s valuable, but the paper should be transparent about how much is new annotation vs relabeling/retiming, and quantify human effort & agreement.

•	Mixed evaluation formats: retro-memory remains MCQ, while other parts use open-ended judged by Qwen2.5-72B. This mixture complicates cross-task comparability and may inherit LLM-judge biases (version drift, style sensitivity).

•	Res Acc metric under-specified: scoring “within a window” ignores latency error magnitude, early/late asymmetry, and false positives (spurious triggers). For stream narration, quality/coverage metrics (recall@IoU, redundancy, hallucination) are not detailed.

•	Sampling & budget fairness: per-model frame budgets differ (e.g., 50 frames for GPT-4o vs 16 for several open-source; 4 fps for “native online”), and the adapted wrapper uses 1 fps windows + 16 memory slots. Without token/pixel budget normalization or policy ablations (uniform vs shot/motion-aware), comparisons may be confounded.

•	Memory module specificity: “nearest-neighbor averaging” over 16 slots is only lightly described. How are keys built, how is compression performed, what is the retention/refresh policy, and what’s the context token cost? Gains could stem from more tokens, not better memory.

•	Quality control dependence on LLMs: LLMs filter “answerable-without-video” items and help synthesize anticipation questions/distractors, which can inject generator artifacts and style priors. Human audit/IAA and leakage checks are not quantified.

•	Forgetting-curve confounds: longer recall intervals may correlate with different content types/difficulties. Matching controls (same target type at different lags) are not described.

**Questions:**

* Please detail the compression, slot update, and NN averaging mechanics, and report ablations on #slots, compression ratio, prompt format, and context tokens to separate memory design from context size.

* How stable are results under alternative judges (e.g., Claude, Gemini) and prompts? For Pro-Anticipation, can you report latency error distributions (mean absolute time error), early/late penalties, and false-trigger rates in addition to Res Acc?

---

> ### Author Response · Authors · 2025-11-25
>
> We would like to thank Reviewer L8cs for their detailed and insightful comments. Your feedback has been invaluable in helping us refine the paper. Below, we provide responses to the comments and suggestions you made. We have also included additional details and clarifications where necessary.
>
> ---
> ### **W1: Data novelty & provenance**
>
> Thank you for your observation on the data novelty and provenance. Indeed, the core innovation of our work lies in adapting existing datasets (Vript-RR, LVBench, LongVideoBench, Ego4D, QVHighlights) into an online interactive protocol. To clarify, around 39% of the annotations are newly constructed, while 61% come from re-labeled and re-timed data. We employed strict data processing procedures, including rule-based filtering followed by human review to ensure the quality of the data. The human effort involved in data processing, annotation, and verification spanned approximately two month, with 96% inter-annotator agreement on the final dataset.
>
> ---
> ### **W2 & W3-2 & Q2-1: Metric Design of Proactive Anticipation**
>
> Thank you for your insightful questions. You’ve pinpointed a key challenge in our evaluation design. The main distinction of the **Proactive Anticipation** task, compared to traditional multiple-choice questions (MCQs), is that the model is required to actively select the timing of its response. Besides the accuracy of the answer, the timeliness of the response is also considered. Additionally, the model's answer should be as unconstrained as possible and given with minimal hints; otherwise, the difficulty of selecting the correct time to answer would be significantly reduced, making it essentially a regular open-ended question. These factors together make the design of a suitable evaluation metric challenging.
>
> First, regarding the issue of **LLM bias**, we addressed it by using **multiple LLMs** to score the responses and comparing the results. The consistency across different LLMs demonstrates the robustness of our evaluation and ensures that the scoring does not overly depend on a single model. Additionally, using open-source models prevents issues such as **weight changes** or **API failures**. We further optimized the scoring process by using **yes/no checklist queries**, which are generated based on the video annotations. This fine-grained breakdown provides more accurate scores by indicating how many queries the predicted outputs successfully recalled. This simplified question format makes it easier to provide fair evaluations that are not influenced by any specific LLM's biases. In the future, we plan to release the checklist queries and prompts used in the evaluation process to ensure transparency and reproducibility.
>
> We also conducted **ablation studies** on the **prompting** strategies, testing different levels of strictness. The results showed good consistency across these prompt variations, which adds confidence to the robustness of our evaluation procedure.
>
> **Table 1: Ablation experiments on metrics of Proactive Anticipation across different models.**
>
> | Judge / Prompt                | VideoChat2-HD | InternVL2.5 | LLaVA-Video | VideoChat-Flash |
> |--------------------------------|---------------|-------------|-------------|-----------------|
> | Qwen2.5-72B (original)         | 4.25          | 4.09        | 5.72        | 5.64            |
> | GPT-4o                         | 3.89          | 3.74        | 5.51        | 5.32            |
> | Qwen2.5-72B + checklist query  | 3.30          | 3.26        | 5.86        | 5.66            |
> | GPT-4o + checklist query       | 3.21          | 3.13        | 5.74        | 5.57            |
> | Qwen2.5-72B strict             | 0.74          | 0.74        | 1.72        | 1.31            |
> | Qwen2.5-72B loose              | 7.20          | 7.52        | 11.45       | 10.38           |

---

> ### Author Response · Authors · 2025-11-25
>
> ### **W3-1 & Q2-2: Awaiting Metrics and Analysis**
>
> Thank you for your valuable suggestions. In response, we have expanded the evaluation metrics, as shown in **Table 2**. The metrics now include:
>
> 1. **Absolute Time Error**: This measures the average absolute error between the response time and the event start time.
> 2. **Time Error Score**: This score is designed in segments, reflecting human subjective tolerance for time errors in Proactive Anticipation tasks. Specifically, it accepts small timing errors, penalizes early responses, and reluctantly accepts late responses.
> 3. **Trigger Ratio**: The proportion of times the model makes its own judgment and generates a response during video interactions.
> 4. **False-Trigger Ratio**: The proportion of responses generated before the event occurs.
>
> The **absolute time error** and **time error score** metrics have shown good consistency across models. Regarding the **trigger-related metrics**, **VideoChat2-HD** performed relatively poorly in triggering decisions. This is because it uses an earlier-stage LLM that struggles to follow instructions well and make timely decisions without training. As a result, its **trigger ratio** is lower. In contrast, **InternVL2.5** exhibited overly confident triggering, resulting in high values for both **trigger ratio** and **false-trigger ratio**.
>
> Additionally, it's important to note that not all videos are answered in a single pass—some questions require further responses or predictions. Since most videos are long, the focus of the test was on long-term interactions and responses. This means that the actual errors in timing will be much larger than the window width, and metric misjudgments are not likely to significantly affect the overall evaluation.
>
> **Table 2: Performance Metrics for Proactive Anticipation Task**
>
> | Model            | Abs Time Error | Time Error Score | Trigger Ratio (%) | False-Trigger Ratio (%) |
> |------------------|----------------|------------------|-------------------|-------------------------|
> | VideoChat2-HD    | 171.5          | 4.51             | 34.67             | 13.13                   |
> | InternVL2.5      | 134.3          | 17.95            | 82.32             | 25.11                   |
> | LLava-Video      | 65.5           | 19.50            | 75.49             | 19.42                   |
> | VideoChat-Flash  | 40.8           | 20.24            | 72.61             | 22.15                   |
>
> ---
> ### **W4 & W5: Memory Module Ablations**
>
> Your concern about fairness regarding model sampling budget and the implementation details of the memory module is highly pertinent, and these factors were primary considerations in our design.
>
> For closed-source models, we tested them via black-box API calls by uploading video files, which inherently restricts our ability to flexibly adjust their internal design. We clarify the frame rate/budget definition for these models: the standard configuration for GPT-4o is 64 frames per query, and for Gemini 1.5 Pro it is 1 FPS. For open-source video understanding models, we maintained consistency with their official code's basic input configuration, regardless of their specific image encoder or connector structure.
>
> For the long-term memory (LTM) module, we strived to ensure that both the number of memory slots ($M$) and the total LTM token count were aligned across adapted models for a fair comparison. Since the Q-Former structure in VideoChat2-HD makes direct token count alignment challenging, we conducted an additional experiment: we expanded the number of memory slots for VideoChat2-HD by a factor of 1.5 (from $M$ to $1.5M$) to precisely match its total LTM token count with the other models. The specific details on token counts are summarized in the table below (further ablations are provided in our response to Q1).
>
> **Table3: Details on Token Alignment for Open-Source MLLMs. *F* represents the number of video frames used for the Short-Term Memory input window. *M* represents the number of Long-Term Memory slots used by the model.**
> | | Visual Feature Size | Connector Type | Short-Term Token Count | Long-Term Token Count |
> | :--- | :--- | :--- | :--- | :--- |
> | VideoChat2-HD | [$F$, 196, 1024] | Q-Former | [96, 4096] | [$M \times 96, 4096$] $\rightarrow$ [**$1.5M \times 96, 4096$**] |
> | InternVL2.5 | [$F$, 256, 1024] | MLP | [$F \times 256, 4096$] | [$M \times 144, 4096$] |
> | LLava-Video | [$F$, 196, 1024] | MLP | [$F \times 196, 3584$] | [$M \times 144, 3584$] |
> | VideoChat-Flash | [$F$/4, 784, 1024] | ToMe + MLP | [$F$/4 $\times 64, 3584$] | [$M \times 144, 3584$] |

---

> ### Author Response · Authors · 2025-11-25
>
> ### **W6: LLM Filtering**
>
> Thank you for your valuable feedback. Our intention with the **LLM filtering** process was to prevent any biases or preferences that the LLM might introduce, which could easily lead to the model inferring the correct answers. We aimed to reduce these biases during the filtering and question generation process. Despite our best efforts, we acknowledge that some stylistic influences may still be present.
>
> To mitigate this, we iteratively modified the generation prompts and templates to ensure that the generated questions were more aligned with the video content, avoiding low-quality options that could deviate too far from the video context. After generating the questions, we manually verified the data, and approximately 90% of the questions were retained after this verification step. This process ensures that the questions used for evaluation remain high-quality and consistent with the video's context.
>
> ---
>
> ### **W7: Forgetting Curve**
>
> Regarding the **forgetting curve** experiment, we used the same video and question content, with the only change being the **timing** of the questions. This allowed us to assess the model's ability to retain information over time without altering the video or question itself. The recall abilities were tested at different time intervals to evaluate how the model's performance varied with the passage of time.

---

> ### Author Response · Authors · 2025-11-25
>
> ### **Q1: Ablation Study on Memory Module, Compression, Prompt, and Window-Size**
>
> Thank you for your thoughtful comments. In response, we conducted ablation experiments to assess the impact of the **memory slots**, **compression ratio**, **prompt**, and **window size** on model performance. All other settings were kept constant. And the nearest-neighbor averaging strategy used in memory compression was shown in Appendix A.3—you can refer to it for details.
>
> ---
> #### **Memory Slots**
> For the **memory slots** ablation, we used **VideoChat2** as the model of choice. As the number of memory slots increased, we observed a consistent improvement in model performance.
>
>
> **Table 4: Performance of VideoChat2 with Varying Memory Slots**
>
> | Slots Num | Short | Middle | Long | Very Long | Avg |
> |-----------|-------|--------|------|-----------|-----|
> | 8         | 34.53 | 35.36  | 32.32| 30.39     | 33.15|
> | 16        | 34.53 | 36.46  | 33.15| 32.60     | 34.19|
> | 24        | 34.25 | 37.29  | 34.53| 32.87     | 34.74|
> | 32        | 35.91 | 37.02  | 34.53| 31.77     | 34.81|
> | 48        | 35.08 | 37.85  | 35.36| 33.15     | 35.36|
>
> ---
> #### **Compression Ratio**
> For the **compression ratio**, we conducted experiments using the **LLava-Video** model, which employs a **ViT-connector-LLM** structure, making it easier to assess the effect of compression. As the compression size increased, model performance continued to improve.
>
> **Table 5: Performance of LLava-Video with Varying Compressed Slot Sizes**
>
> | Compressed Size of Each Slot | Short | Middle | Long | Very Long | Avg |
> |------------------------------|-------|--------|------|-----------|-----|
> | 3                            | 44.20 | 43.65  | 37.02| 37.85     | 40.68|
> | 4                            | 48.07 | 45.86  | 40.51| 37.57     | 43.00|
> | 5                            | 48.90 | 48.34  | 43.65| 38.40     | 44.82|
> | 6                            | 48.43 | 50.79  | 43.22| 38.80     | 45.29|
> | 9                            | 48.62 | 50.28  | 42.54| 40.05     | 45.37|
>
> ---
>
> #### **Prompt**
> We also conducted an ablation study on the **prompt**. The **system prompt** is used to formally guide the model to carefully observe the video and answer the questions. The **LongShort-Clip prompt** provides guidance for the video segment corresponding to memory. We tested different prompt formats and observed how they impacted performance across models. The absence of prompt will result in a certain degree of performance degradation.
>
> **Table 6: Performance with Different Prompts Across Models**
>
> | System prompt | LongShort-Clip prompt | VideoChat2-HD | InternVL2.5 | LLaVA-Video | VideoChat-Flash |
> |---------------|-----------------------|---------------|-------------|-------------|-----------------|
> | √             | √                     | 34.19         | 36.33       | 40.68       | 44.34           |
> | √             |                       | 33.36         | 35.98       | 39.23       | 41.71           |
> |               | √                     | 33.84         | 36.26       | 39.51       | 41.98           |
>
> ---
> #### **Window Size**
> We also explored the effect of **window size**, or **context length**, on model performance. As expected, model performance increased with larger window sizes up to a point, after which performance began to plateau and even decrease slightly. This shows that beyond a certain window size, the model may not benefit from additional context.
>
> These results showed that changing the memory slot size provided more stable performance gains while keeping token counts manageable. Compared to Tables 4 and 5, it can be observed that modifying the number of memory slots and the compression rate led to more consistent improvements.
>
> **Table 7: Performance of VideoChat2 with Varying Window Sizes**
>
> | Window-Size | Short | Middle | Long | Very Long | Avg |
> |-------------|-------|--------|------|-----------|-----|
> | 8           | 36.46 | 35.36  | 34.53| 30.39     | 34.19|
> | 16          | 34.53 | 36.46  | 33.15| 32.60     | 34.19|
> | 24          | 35.08 | 36.19  | 35.08| 33.43     | 34.94|
> | 32          | 34.81 | 35.36  | 35.91| 31.77     | 34.39|
> | 48          | 35.36 | 37.85  | 32.79| 30.50     | 34.07|
> | 64          | 36.74 | 33.70  | 34.81| 30.66     | 33.98|
>
> **Table 8: Performance of Llava-Video with Varying Window Sizes**
>
> | Window-Size | Short | Middle | Long | Very Long | Avg  |
> |-------------|-------|--------|------|-----------|------|
> | 8           | 43.65 | 43.37  | 35.64| 32.87     | 38.88|
> | 16          | 44.20 | 43.65  | 37.02| 37.85     | 40.68|
> | 24          | 45.30 | 45.86  | 38.67| 38.12     | 41.99|
> | 32          | 46.41 | 46.96  | 40.82| 40.06     | 43.56|
> | 48          | 46.96 | 46.69  | 40.06| 38.67     | 43.09|
> | 64          | 48.34 | 47.51  | 39.23| 35.82     | 43.13|

---

> > ### Comment · Reviewer_L8cs · 2025-11-27
> > **Thanks to the authors**
> >
> > I appreciate the author's efforts and detailed response.
> >
> > Thanks for including more analysis and experiments that polish your paper further, e.g., ASR inclusion, memory ablations, and varying window sizes.
> >
> > After carefully reading the rebuttal and the other reviewers' comments, I was willing to increase my score to 8, rather than 6.
> > But I noticed that the data has not been released yet.
> > Therefore, please release the data as soon as possible. You can upload it anonymously as supplementary materials or through an anonymous link.

---

> ### Author Response · Authors · 2025-11-28
> **Acknowledgment & Data Upload**
>
> Thank you for your thoughtful feedback and for carefully reviewing the additional experiments. Your suggestions were truly helpful for improving the paper. I also appreciate your patience and professionalism in going through all the discussions in detail.
>
> Following your recommendation, I have now uploaded the annotated data as supplementary materials to ensure that everyone can access and verify it during the review process.
>
> Thank you again for your positive and constructive attitude throughout. Please feel free to let me know if there are any further questions or concerns.

---

### Author Response · Authors · 2025-12-03
**Summary of Rebuttal**

First, we sincerely thank all reviewers for their time and valuable feedback, which greatly helped us improve the paper. Below, we summarize the rebuttal.

---
### **Restating Core Innovations**
- **Streamed Input & Interactive Tasks**: We define a **real-time interaction framework** with continuous video input and proactive, flexible responses to user queries.
- **Evaluation Metrics**: We introduce metrics that jointly assess response **accuracy and timeliness** in interactive settings.
- **Active Response Capability**: We present a simple, transferable method to convert offline video models into online ones, enabling **active response**. This providing both a baseline for future work and validation for our metrics.
---
### **Addressing Reviewer Comments**
1. **Memory Module Design & Ablation**
- Reviewer `L8cs` and `HL6Z` requested some details and ablations. We provide extensive ablations across several dimensions showing that compressed memory consistently improves performance, not just from added computation.
- Reviewer `ya2u` questioned about adaptive eviction or learned compression. We affirmatively address this point, confirming the feasibility and benefits of such enhancements.

2. **Evaluation Metrics for Proactive Anticipation (Instant Task)**
- Reviewers `L8cs` and `9Ctw` encouraged richer timing evaluation. In response, we have introduced complementary metrics, including mean absolute time error, early/late penalties, and false-trigger rates.
- Reviewer `HL6Z` asked the definition of response window and suggested adding uncertainty metrics. We clarify it and agree that modeling uncertainty is valuable for ambiguous events.

3. **Evaluation Metric for Proactive Anticipation (Streaming Task)**
- Reviewer `L8cs` suggested evaluating judge stability across models. We evaluated responses using multiple LLMs as judges.
- Reviewer `9Ctw` encouraged more detailed reporting on prompt design. We conducted experiments with prompts of varying strictness and report the results.
- Reviewer `ya2u` recommended reducing dependence on a single judge. We reformulated the evaluation into a set of binary correctness judgments, significantly enhancing objectivity.
Collectively, these evaluations demonstrate strong consistency, validating the reliability of our open-ended assessment protocol.

4. **Data Provenance and Generation**
- Reviewer `L8cs` called for clear quantification of data generation. We clarify that substantial human effort was invested in designing the questions and interactive formats.
- Reviewers `L8cs` and `9Ctw` suggested ensuring robustness in synthetic data generation. We refined the generation and filtering pipeline through multiple validation rounds to ensure quality.
- Reviewer `ya2u` recommended sharing generation artifacts for transparency. We have added representative examples in the appendix of the revised paper.

5. **Benchmark Scope and Domain Coverage**
Reviewer `HL6Z` suggested expanding to more dynamic or high-stakes scenarios. Reviewer `ya2u` encouraged broader domain coverage beyond current sources, like robotics, meetings, and sports. We clarify that our benchmark **already covers** the mentioned scenarios, demonstrating strong generalizability.

6. **Fairness in Model Comparison and Resource Budgets**
We corrected the frame count description for closed-source models and updated the paper. In response to reviewers `L8cs`, `9Ctw`, and `ya2u`’s suggestion on fair comparison, we include a dedicated table aligning computational budgets across models, already accounted for in our initial experimental design.

7. **Absence of Audio Modality**
Reviewers `HL6Z` and `ya2u` highlighted audio’s importance for real-time interaction. We incorporated ASR transcripts with visual-text inputs, yielding measurable gains that confirm the value of audio integration.

8. **Future Directions**
Reviewer `ya2u` uniquely suggested forward-looking improvements to enhance community impact:
- Testing under realistic imperfections (e.g., timestamp jitter, dropped frames, ASR lag);
- Simulating safety-critical behaviors for embodied agents (e.g., intervention thresholds, recovery from errors);
- Supporting curriculum learning via psychometric difficulty analysis.
We plan to refine the benchmark accordingly, with a focus on real-world applicability.

---
### **Summary of Further Feedback**
Reviewer `L8cs` expressed willingness to raise their score **from 6 to 8** and, per their request, we have uploaded the data. The insightful comments greatly strengthened the benchmark’s rigor and the method's validation.

Reviewer `ya2u` gave a high-confidence but somewhat ambiguous response, suggesting many valuable future directions. However, `ya2u` seemed to focus mainly on the memory module, part of our last innovation, possibly overlooking the broader significance of online video interaction. Unfortunately, we couldn’t further discuss to clarify or address any potential misunderstanding.

---

### Meta-Review · Area_Chair_NdML · 2026-01-08

**Summary:**

This paper proposes a new benchmark to evaluate online video understanding.

The reviews were mixed across the four reviewers, with two 4's and two 6's given initially.  Most of the concerns were related to clarification of various experimental settings, adding more ablation studies and settings, adding modalities (sound), etc.  These were almost all addressed in the rebuttal.

**Reviewer Concerns:**

Most of the questions and concerns from each reviewer were clarification-based, or asking about alternative experimental settings.  The authors addressed almost all of these in one form or another.

There are no remaining concerns from what I can see.

**Reviewer Scores:**

L8cs - gave a 6 originally with low confidence (2). I think this reviewer would've raised their score.
9Ctw - gave a 6 originally, with mid-level of confidence (3)
HLZ6 - gave a 4 originally, with mid-level of confidence (3).
ya2u - gave a 4 originally, with high confidence (5).

I think almost all of the reviewers would've raised their scores.  Most of the questions and concerns from each reviewer were clarification-based, or asking about alternative experimental settings.  The authors addressed almost all of these in one form or another.

---

### Decision · Program_Chairs · 2026-01-26

Accept (Poster)